# Gating intermediates reveal inhibitory role of the voltage sensor in a cyclic nucleotide-modulated ion channel

Xiaolong Gao[1,5], Philipp A. M. Schmidpeter [1,5], Vladimir Berka[2], Ryan J. Durham[2], Chen Fan[1,4], Vasanthi Jayaraman[2] & Crina M. Nimigean [1,3] ✉

Understanding how ion channels gate is important for elucidating their physiological roles and targeting them in pathophysiological states. Here, we used SthK, a cyclic nucleotide-modulated channel from *Spirochaeta thermophila*, to define a ligand-gating trajectory that includes multiple on-pathway intermediates. cAMP is a poor partial agonist for SthK and depolarization increases SthK activity. Tuning the energy landscape by gain-of-function mutations in the voltage sensor domain (VSD) allowed us to capture multiple intermediates along the ligand-activation pathway, highlighting the allosteric linkage between VSD, cyclic nucleotide-binding (CNBD) and pore domains. Small, lateral displacements of the VSD S4 segment were necessary to open the intracellular gate, pointing to an inhibitory VSD at rest. We propose that in wild-type SthK, depolarization leads to such VSD displacements resulting in release of inhibition. In summary, we report conformational transitions along the activation pathway that reveal allosteric couplings between key sites integrating to open the intracellular gate.

Ion channels are responsible for electrical signaling across cellular membranes and, as such, are involved in diverse physiological processes including vision and taste sensation, touch and temperature perception, pain, neuronal firing, autonomous pacemaking, apoptosis[1–5]. Ion flux occurs through the channel pore which can be linked to one or more stimulus-specific, regulatory domains that tightly control the opening and closing (gating) of the channel according to the physiological need. Gating information is allosterically transmitted from the regulatory domains to the pore domain and can involve large-scale conformational rearrangements of the entire protein. Such conformational transitions are difficult to resolve structurally and even resolving the end states (resting or closed state, and active or open state) is not always possible, not to mention that end states alone are not enough to properly define the activation pathway. Therefore, gating intermediates are crucial to understanding

the energetics of ion channel regulation, the development of diseases, and for the design of specific therapeutics to treat channelopathies.

High-resolution cryogenic electron microscopy (cryo-EM)[6] has led to the determination of many new ion channel structures in distinct functional states. Plunge freezing samples for cryo-EM ideally should preserve gating intermediates and provide a picture very similar to the state distribution in solution. However, such intermediates are energetically unstable and hence only transiently populated and not abundantly present, as evidenced by the scarcity of structures for such gating intermediates[7–10]. Closed and open structures of ion channels that are gated by ligands, are generally obtained in the absence and presence of full agonists, ligands that open these channels efficiently and, thus, bias the equilibrium strongly towards open states. On the other hand, less efficient, partial agonists may favor the relative stabilization of intermediate states, a strategy we employed here.

[1]Department of Anesthesiology, Weill Cornell Medical College, 1300 York Avenue, New York, NY 10065, USA. [2]Department of Biochemistry and Molecular Biology, University of Texas Health Science Center, Houston, TX, USA. [3]Department of Physiology and Biophysics, Weill Cornell Medical College, 1300 York Avenue, New York, NY 10065, USA. [4]Present address: Science for Life Laboratory, Department of Biochemistry and Biophysics, Stockholm University, Solna, Sweden. [5]These authors contributed equally: Xiaolong Gao, Philipp A. M. Schmidpeter. ✉e-mail: crn2002@med.cornell.edu

Cyclic nucleotide-gated (CNG) ion channels are activated by cyclic nucleotides and only slightly modulated by voltage, although the mechanism is not yet understood[1,11]. Physiologically, they are key players in visual and olfactory signaling cascades and dysfunction of these channels leads to channelopathies like achromatopsia or anosmia[1,12,13]. CNG channels are tetrameric channels where each subunit consists of a voltage sensing domain (VSD, S1–S4), a pore domain (S5-S6), and a cytosolic cyclic nucleotide binding domain (CNBD) that is linked to the pore domain via a C-linker comprised of four α helices (Fig. 1a). Depending on the isoform, these channels are differentially activated by cyclic nucleotides (cAMP or cGMP). Structures of both open and closed *C. elegans* TAX-4 channels[14,15] and homomeric human rod CNGA1[16] demonstrated that these eukaryotic channels undergo similar conformational changes when end states are compared, echoing previous functional and structural studies of CNG channels[17–23].

Cyclic nucleotides are very efficient in opening eukaryotic CNG channels, making it difficult to detect gating intermediates[1]. In contrast, cAMP is only a partial agonist for SthK, a prokaryotic CNG channel[24–27]. In agreement with this, cryo-EM structures of WT SthK yielded only closed states even in the presence of saturating cAMP[28]. Meanwhile, membrane depolarization increases cAMP-induced channel activity, suggesting that voltage modulates the agonist status of cAMP in SthK, similar to its action on other CNG channels, although the mechanism is not known[1,24–26,29].

Here we take advantage of the partial agonist nature of cAMP for SthK channels and the synergistic activation by ligands and voltage, to capture gating intermediates along the activation pathway of SthK and to understand how voltage modulates channel activity. To increase the efficacy of cAMP for SthK without interfering with the ligand gating pathway, we screened for gain-of-function mutants of SthK that mimic depolarization. SthK has a voltage-sensor like domain (S1–S4) with only 4 basic residues, compared to 7 to 9 in other voltage-gated ion channels likely contributing to the smaller gating charge (0.8 compared with 4 in Shaker and >3 in HCN, from Boltzmann fits of $P_o$ vs. voltage plots)[24,30–32]. Cryo-EM structures of VSD mutants with increased activity yielded cAMP-bound closed states (identical to apo and cAMP-bound WT SthK), as well as four additional states: one gating intermediate and 3 open states with different degrees of opening.

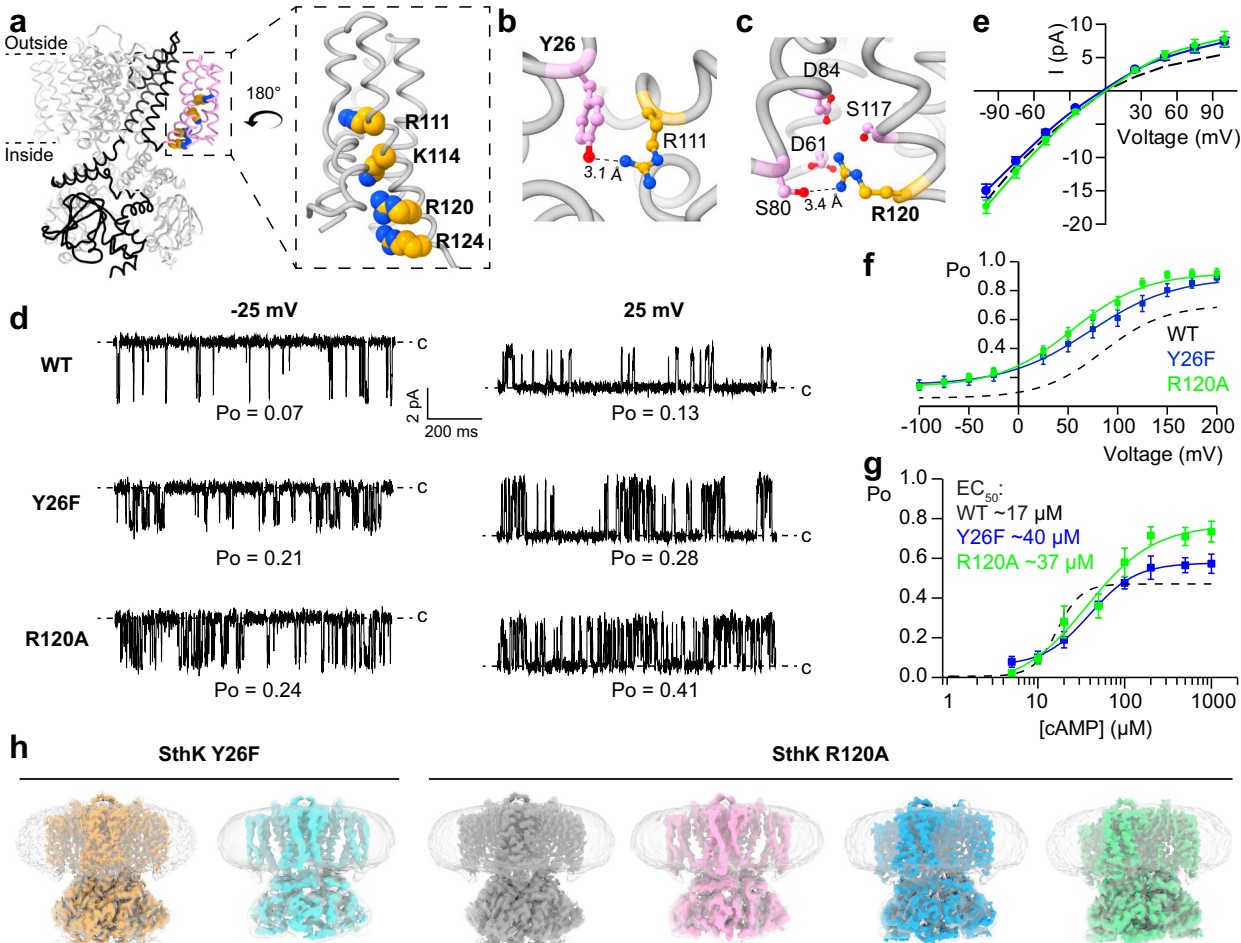

**Fig. 1 | Identification and characterization of gain-of-function VSD mutants.**
**a** Structural overview of SthK (PDB: 6cju), one subunit is highlighted in black with the voltage sensing domain in pink and conserved, basic residues on the S4 helix as spheres. Inset highlights the voltage sensor with the positively charged residues indicated. **b** Interactions between Y26 on the S1 helix (pink) and R111 on the S4 helix (yellow). **c** Interaction between R120 on the S4 helix (yellow) and several surrounding residues on S2 and S3 (pink). **d** Single channel recordings of WT SthK, SthK Y26F, and SthK R120A at −25 mV and +25 mV in the presence of 200 μM cAMP. Closed levels are indicated by dashed lines. **e** Single channel current amplitude as a function of voltage (*I/V*) for WT SthK (dashed, from ref. 24), SthK Y26F (blue), and SthK R120A (green) in the presence of 200 μM cAMP from recordings as in (**d**). **f** Open probability in the presence of 200 μM cAMP as function of the applied voltage. Colors as in (**e**). Lines represent fits according to Eq. 1. **g** Open probability at +100 mV as function of the cAMP concentration. Colors as in (**e**). Lines represent fits according to Eq. 2. All fitted parameters from electrophysiological recordings are summarized in Supplementary Table 1. **h** Cryo-EM maps of the different channel conformations obtained for SthK Y26F and SthK R120A in lipid nanodiscs. Unsharpened maps and density for nanodiscs, are in gray/transparent. Different colors highlight the different structures. Symbols in **e**–**g** represent mean ± SEM. *n* = 3–20 independent experiments for all data points. Dashed lines in **e**–**g** represent WT SthK data from ref. 24. Source data are available as a Source Data file.

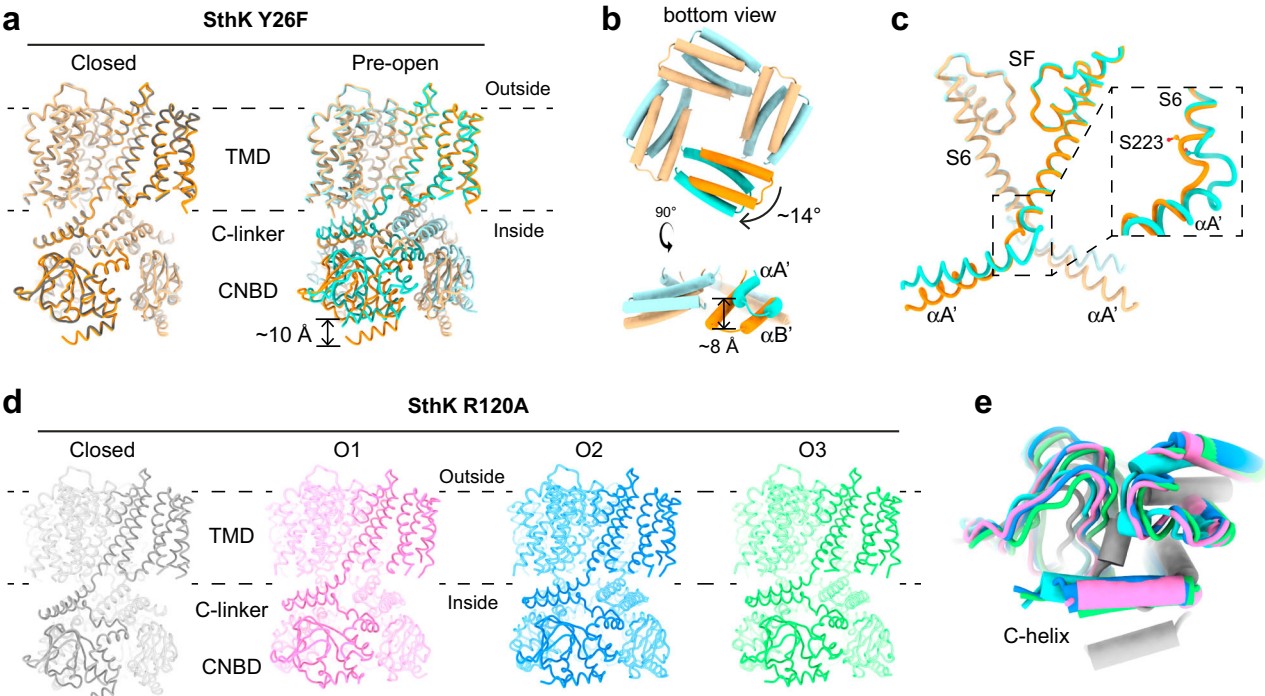

**Fig. 2 | Structures of different conformations for SthK Y26F and SthK R120A.** **a** Overlay of closed SthK Y26F (orange, left) with closed WT SthK (gray, PDB: 6cju) and with activated, pre-open SthK Y26F (cyan, right). **b** C-linker movements between closed (orange) and activated pre-open (cyan) SthK Y26F. **c** Opening of activated, pre-open SthK Y26F at the hinge between S6 and C-linker (colors as in **a**, **b**, an opposing subunit is shown in transparent). **d** Atomic models for SthK R120A: closed state (gray), open state 1 (O1, pink), open state 2 (O2, blue), open state 3 (O3, green). Three subunits of each model are transparent for clarity. **e** Overlay of one CNBD from pre-open SthK Y26F (cyan) and SthK R120A closed state (gray), O1 (pink), O2 (blue), O3 (green) showing conserved intra-CNBD movements.

These substates reveal key transitions between channel conformations that highlight the allosteric coupling between the voltage-sensor and ligand binding domains, and how they contribute to gate opening. Single-molecule fluorescence resonance energy transfer (sm-FRET) measurements indicated that the intermediates are a common feature of SthK, although there is increased dynamics in the mutant channels. Our findings provide a unique framework to describe CNG channel activation by ligands, the modulatory effect of voltage, and provide insights into the mechanism of partial agonism in these channels.

## Results

### Voltage sensor mutations that increase channel activity

cAMP is a poor partial agonist for SthK, and cryo-EM structures of WT SthK led to only closed states even in the presence of excess cAMP[28]. In order to capture additional states along the activation pathway, we sought to increase the efficacy with which cAMP opens SthK channels. Our approach for increasing the efficacy while minimally interfering with the ligand-gating pathway and at the same time learning about the voltage sensing mechanism, was to introduce small perturbations to the voltage sensor domain (VSD) via mutation, and screen for channels with higher activity (Fig. 1 d–g and Supplementary Fig. 1). Based on the structure of SthK at rest[28], we identified candidate residues in the VSD for mutation designed to disrupt the interaction network that holds the S4 helix in its resting state, with the idea that S4 needs to move for the channel to open (Fig. 1a–c and Supplementary Fig. 1). Two mutants, Y26F and R120A, showed good expression levels and the required increased activity, which allowed us to perform further studies (Fig. 1d–g and Supplementary Figs. 1, 3a and 5a).

Tyr26 is located on the S1 helix, and its hydroxyl group is coordinated by the guanidine group of Arg111, a conserved arginine in the S4 helix and presumably important for voltage sensing in SthK (Fig. 1b). Removal of the hydroxyl group, as in SthK Y26F, abolishes this interaction, which can conceivably facilitate S4 movement in the

membrane. Similarly, the sidechain of Arg120, another basic residue on the S4 helix, points into an electronegative pocket formed by Asp61, Ser80, Asp84, and Ser117 on the S2 and S3 helices (Fig. 1c). Removal of the positive charge of Arg120 would also weaken interactions between S4 and the neighboring helices, facilitating VSD movements in the membrane.

Both SthK Y26F and SthK R120A are activated by cAMP similar to WT SthK (Fig. 1d) but the open probability (Po) is increased across all voltages (from −100 to +200 mV), at saturating cAMP (Fig. 1f). Importantly, at 0 mV, the condition on the cryo-EM grid, the extrapolated Po of both mutant channels is ~0.25, more than doubling the activity previously observed for WT SthK (~0.1)[24,28]. The voltage dependence is similar, and the observed gating charges range from 0.6 to 0.7 in SthK Y26F and SthK R120A, respectively (compared to 0.8 for WT SthK, Supplementary Table 1 [24]). Both mutants show inwardly rectifying single channel amplitudes in the current–voltage (I/V) relation, characteristic for SthK (Fig. 1d, e). Activation of both mutant SthK channels is strictly cAMP-dependent (Fig. 1g and Supplementary Fig. 8d) with $EC_{50}$ values of 40 μM for SthK Y26F and 37 μM for SthK R120A at 100 mV (cf. 17 μM for WT SthK)[24] and no activity was observed in the absence of cAMP in either mutant. The Hill coefficients required to fit the data in Fig. 1g are smaller for the mutants compared to WT SthK (Supplementary Table 1). While this may indicate a change in cooperativity of ligand binding, more investigation is needed on this topic. Importantly, the maximal open probabilities of the mutants are higher than those for WT SthK (with SthK R120A slightly higher than SthK Y26F), indicating that we succeeded in turning cAMP into a more effective agonist (Fig. 1d–g and Supplementary Table 1). This supports our hypothesis that the VSD can modulate the agonist status of cAMP.

### SthK Y26F reveals a closed but activated conformation

To gain structural insights into the effects of the two VSD mutations on the conformational landscape of SthK we reconstituted both proteins

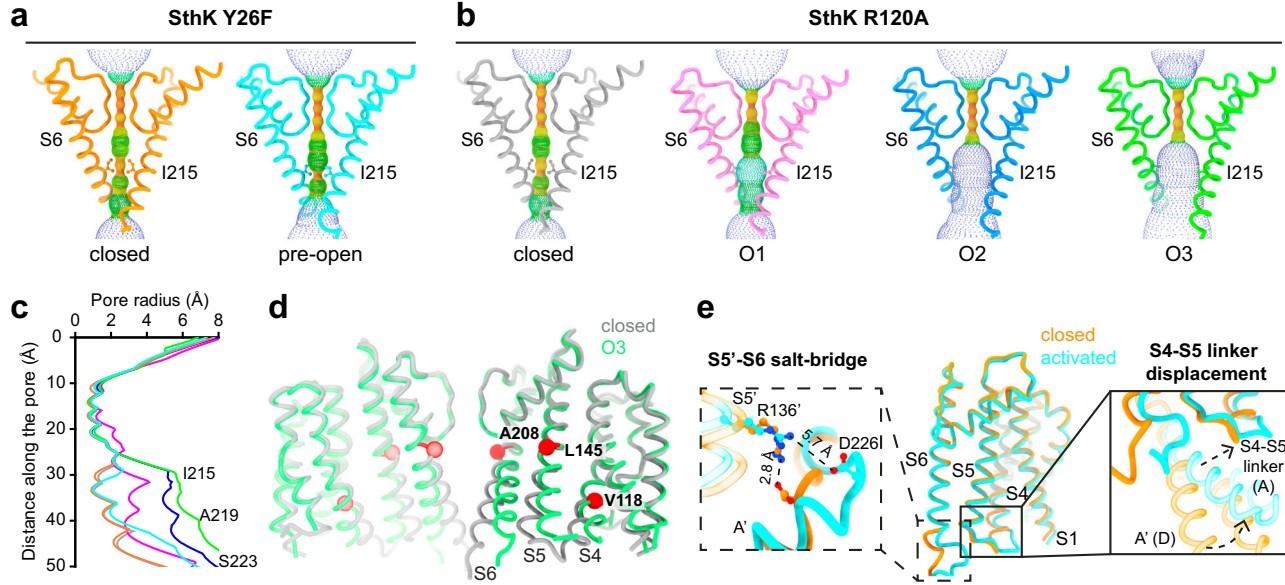

**Fig. 3 | Gating intermediates display gradually increased pore openings.**
**a**, **b** Show the pore helices, selectivity filters, and S6 pore lining helices from two opposing subunits (cartoon) and the pore radius (spheres, calculated using HOLE) for SthK Y26F in (**a**) and SthK R120A in (**b**). Ile215, which forms the main constriction, is highlighted and shown as sticks. **c** Plot of the calculated pore radii for all proteins used in this study with colors as in (**a**, **b**). **d** Overlay of SthK R120A closed (gray) and O3 (green) states showing displacements of S4, S5, S6 during channel

opening. Hinge points on each helix, where movements begin, are highlighted (red) and the corresponding residues are labeled. **e** TMD of one subunit of SthK Y26F (closed in orange, pre-open in cyan) is shown. Zoom-ins focus on the Arg136-Asp226 inter-subunit salt-bridge holding the bundle crossing closed (left) and the S4-S5 linker displacement in the pre-open state allowing the C-linker to move upwards closer to the membrane.

into lipid nanodiscs using the same lipid composition as in our functional experiments (DOPC:POPG:Cardiolipin, 5:3:2) for cryo-EM studies. SthK Y26F adopts two conformations in the presence of cAMP (Fig. 1h, Supplementary Fig. 2, 3, Supplementary Table 2). The majority of particles (1112621 particles, ~91%) was in a closed conformation, identical to the closed state of WT SthK (Fig. 2a, RMSD = 0.4 Å) as expected from our functional data (Po at 0 mV is ~0.25). Importantly, the Y26F mutation does not lead to any local or other changes in the channel as gleaned from the identical structures of WT SthK and SthK Y26F in the closed state (Fig. 2a, Supplementary Fig. 6). The second conformation (109647 particles, ~9%) displayed global rearrangements in the cytosolic domains in agreement with the proposed ligand gating movements in both SthK and CNG channels in general (Fig. 2a, b)[17,23,28,33,34]. The CNBD adopts the same structure as the isolated, activated CNBD of SthK solved by X-ray crystallography[35], where the terminal C-helix moves towards the binding pocket by ~10 Å to contact the bound cAMP and to close over it like a lid (Supplementary Fig. 7a). In addition, the entire CNBD translates as a rigid body towards the membrane, accompanied by upwards displacements of the C-linker helices. In particular, helices A' and B' of the C-linker move closer to the membrane by ~8 Å. The C-linker also rotates outwards by 14° relative to its resting position, which leads to a radial expansion of the CNBD ring by ~4 Å (Fig. 2b, e and Supplementary Fig. 7b, c).

While such rearrangements in the ligand binding domain are accompanied by pore opening in CNG channels[14,16], there is little change in the transmembrane region of SthK Y26F (Fig. 2c). The upwards movement of the C-linkers gradually fades out towards the N-terminal end of the A' helix. The rotation of the A' helix, however, exerts strain on the C-terminal end of the S6 pore-lining helix, to which it is connected via a three-residue loop. This strain is accommodated by distending the intracellular end of the pore at the bundle-crossing region, at the level of the pore-lining residue Ser223, which moves away from the pore (Fig. 2c). However, this expansion is limited to the very bottom of the intracellular entry to the pore (the last helical turn of S6) and is insufficient to generate an outwards rotation of S6. Thus,

the lumen above Ser223, which is similar to the central gate found in eukaryotic CNG channels, is not changed as compared to the closed state. Especially, Ile215, which forms the main constriction in SthK and other CNG channels, still pinches the pore sterically shut below the selectivity filter (Fig. 3a, c)[16,28]. The VSD is virtually unchanged from its conformation in the closed state. In summary, the second conformation observed for SthK Y26F features an activated C-linker/CNBD assembly but with a closed pore. We assign this state to an activated, pre-open conformation, where cAMP-binding induced the conformational changes necessary for opening the channel, but an open pore conformation is not sufficiently stabilized to be structurally resolved.

## SthK R120A reveals multiple open states

We next solved the structures of SthK R120A (Supplementary Figs. 4 and 5 and Supplementary Table 3). Again, as predicted, most of the particles were in a closed conformation (799349 particles, ~76 %), virtually identical to that of WT SthK and SthK Y26F (RMSDs between the closed state structures of each mutant with respect to WT SthK are 0.4 and 0.7 Å, respectively, Supplementary Fig. 6). Again, similar to SthK Y26F, the R120A mutation does not lead to any local or other changes in the channel as gleaned from the identical structures of WT SthK and SthK R120A in the closed state (Supplementary Fig. 6).

In contrast to WT SthK and SthK Y26F, but in agreement with the higher open probability of SthK R120A, we also captured three open state conformations for SthK R120A (Fig. 1h). The C-linker/CNBD domains of these three states are all in the activated state, with the C-helix in contact with cAMP, the entire C-linker/CNBD assembly translated towards the membrane, rotated by about 12° with respect to the transmembrane region, and expanded radially with respect to the pore axis. These C-linker/CNBD movements are identical to those observed for pre-open SthK Y26F and other CNG channels (Fig. 2d, e and Supplementary Fig. 7b, c)[14,16,34].

The three additional conformations for SthK R120A all display open pores, albeit to different extents. We call them open states O1, O2, and O3, with O3 being the most open conformation (Fig. 2d and

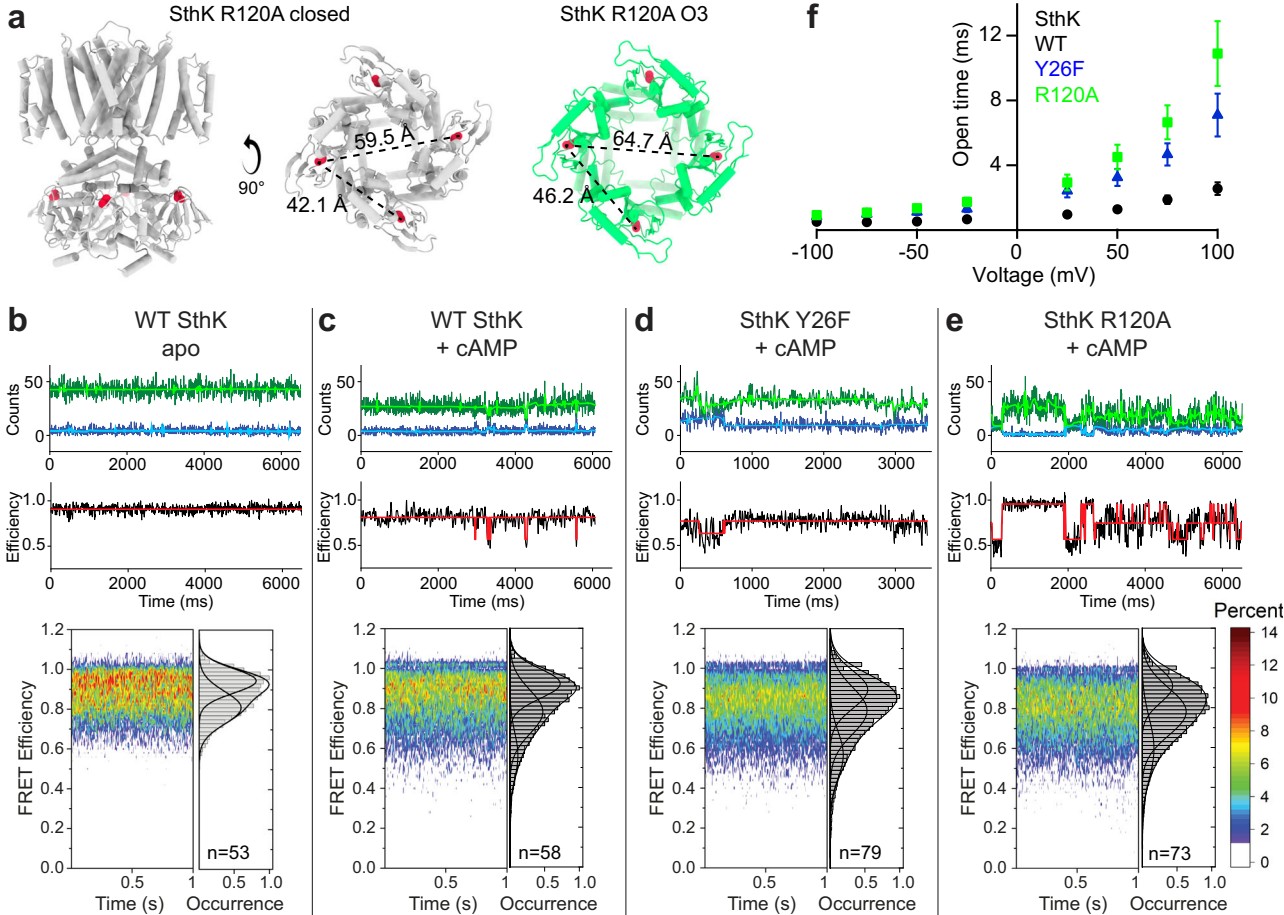

**Fig. 4 | Identification of gating intermediates in WT SthK. a** SthK R120A in closed (gray) and open state (O3, green) are shown with Cys387 for fluorophore labeling highlighted (red). Distances between Cys387 from adjacent and opposing subunits are given for both states. **b–e** Top row: representative sm-FRET donor (blue) and acceptor (green) traces, as well as denoised traces (light blue and light green, respectively). Middle row: observed FRET efficiencies and idealized traces (black and red, respectively). Bottom row: distributions of observed FRET efficiencies displayed as sm-FRET contour plots and cumulative population histograms (gray) generated from the compilation of all analyzed single-molecule traces, with n indicated in the panels. Population histograms were fitted with Gaussian functions and parameters are summarized in Supplementary Table 4. **f** Open dwell times obtained from recordings as in Fig. 1d (see representative histograms in Supplementary Fig. 8f, g) for WT SthK (black), SthK Y26F (blue), and SthK R120A (green) are plotted as function of the applied voltage. Symbols represent mean ± SEM, $n$ = 4–9 independent experiments for all data points. Source data are available as a Source Data file.

Fig. 3b, c; O1: 50037 particles, ~5%; O2: 149792 particles, ~14%; O3: 46629 particles, ~5%). The strain generated by elevating and rotating the C-linker/CNBD domains in SthK R120A is efficiently transmitted to the pore lining S6 helices that are rotated outwards, opening the intracellular bundle crossing region[27]. S6 hinges outwards at Ala208, located right underneath the selectivity filter (Fig. 3d). Along with the S6 backbone translation and rotation away from the pore axis, the side chains of Ile215, which form the main constriction in the closed state similar to the cavity gate in CNG channels, also swing away from the pore, creating large intracellular gate openings (Fig. 3b, c). The gradual increase of the pore radius from O1 to O2 to O3 is due to different endpoints of the S6 helix transitioning towards the final open state, O3, which suggests that O1 and O2 are on-pathway gating intermediates towards a fully open channel (Supplementary Fig. 7d–g).

**VSD positioning allows for different gate expansions**
The CNBD and C-linker regions of activated, pre-open SthK Y26F, and open SthK R120A structures are basically identical and display the same differences when compared to the closed resting SthK structure (Fig. 2e and Supplementary Fig. 7b, c)[28]. Thus, although the cAMP binding-induced conformational changes in the C-linker/CNBD are complete in all these structures, these rearrangements were not sufficient to stabilize SthK Y26F in open states in our samples, while they

were sufficient to do so in SthK R120A. To gain further mechanistic insight into these differences, we closely inspected the VSD positioning relative to the pore in these structures.

In all three open conformations of SthK R120A, the lower two helical turns of the S4 helix of the VSD are displaced outwards by a small kink at Val118, allowing S5 to move, which creates space for the expansion of S6 (Fig. 3d and Supplementary Fig. 7d, e). This lateral displacement is not observed in the closed states, which show identical S4 helices in SthK WT, Y26F, and R120A. The movements of S4 and S5 gradually increase among the three open conformations explaining the gradual increase in pore diameter (O1 < O2 < O3) (Fig. 3c and Supplementary Fig. 7d–g). In contrast, in activated, pre-open SthK Y26F, S4 shows only a minimal displacement which is limited to the last helical turn and does not allow expansion of S6 (Supplementary Fig. 7e). This displacement is, however, accompanied by a repositioning of the S4-S5 loop, which generates the space necessary for the upwards translation of the C-linker/CNBD domains (Fig. 3e). At the same time, the S5 helix in the pre-open state of SthK Y26F is identical to that in the closed state, allowing for only negligeable movements of S6 below the gate level (i.e., below Ile215, Fig. 2c and Fig. 3e). In the closed state, the bundle crossing is stabilized by an inter-subunit salt-bridge between Arg136 on S5 and Asp226 on S6[27]. In the activated, pre-open conformation, although the pore is still closed, this salt-bridge is

broken (Fig. 3e) indicating that the gate is poised to open in this conformation.

In summary, our findings indicate that voltage sensor, especially S4, displacements may allow for more efficient coupling between the conformational rearrangements induced by ligand binding in the CNBD/C-linker and opening of the gate, which enabled us to capture different gating intermediates. This supports the hypothesis that the gain-of-function mutations in the VSD led to channels with different levels of increased mobility of the S4 helices, pointing towards an inhibitory role of VSDs on channel activation. Release of such inhibition to increase SthK channel activity can be achieved upon depolarization that displaces the voltage sensor domains (Fig. 1f)[24]. Therefore, both membrane depolarization and the gain-of-function mutations favor a repositioning of S4 to relieve the inhibition and to allow for more efficient coupling between cAMP-binding and channel opening. This is further supported by the fact that the closed state structures of these mutants are identical to the closed state of WT SthK. In both cases, SthK Y26F and SthK R120A, the mutation did not lead to different protein conformations at rest but altered the relative stabilities between closed and open states.

### sm-FRET confirms gating intermediates in WT SthK

Our results so far revealed four new conformations along the gating pathway of SthK, which we achieved by changing the energetics of the conformational landscape via mutations designed to mimic depolarization. The structure of WT SthK in saturating cAMP yielded only a closed state identical to that of the apo channel[28]. However, cAMP increases the open probability of WT SthK to ~10% at 0 mV which means that conformations other than the resting state are sampled over time in the WT channel but are not sufficiently stable to be captured with cryo-EM (Fig. 1d–f)[24,28].

To investigate if WT SthK samples similar gating intermediates as observed in the mutants, we employed single molecule Fluorescence Resonance Energy Transfer (sm-FRET)[36–40]. Sm-FRET can provide structural and dynamic information about proteins in the form of distance fluctuations derived from FRET efficiencies monitored over time[37,40]. When a FRET donor and acceptor pair is in close proximity, FRET efficiency is high, while when the two moieties move away from each other, FRET efficiency decreases[41]. Unlike cryo-EM but similar to single channel electrophysiology recordings, sm-FRET measurements are performed in solution with a time resolution in the millisecond range. Thus, sm-FRET is ideally suited to study conformational changes between different mutant SthK channels and different liganded states.

We introduced FRET donor–acceptor pairs in SthK via sulfhydryl chemistry on Cys residues. SthK has two Cys residues per monomer (Cys153 in S5, and Cys387 in the CNBD). For sm-FRET measurements we used SthK C153A, which has only one Cys per CNBD for labeling. SthK C153A behaves like WT SthK in single-channel recordings (Supplementary Fig. 8a, d), and we will refer to it from here on as WT. Similarly, sm-FRET experiments on SthK Y26F and SthK R120A were also performed on the background of SthK C153A, and we will refer to them as SthK Y26F and SthK R120A. Although differences are less pronounced in the double-mutants, their open probabilities are still higher than for SthK C153A (Supplementary Fig. 8a, e).

We used Alexa 555 and Alexa 647 as donor-acceptor pair[36–40]. Based on our structures, Cys387 of adjacent subunits are within ~42 Å in the resting state and ~46 Å in the open state ($C_\alpha$ to $C_\alpha$ distance, Fig. 4a). Both distances would thus result in greater than 60% FRET efficiency. However, with fluorophores attached to Cys residues, these distances likely are larger and can show lower FRET efficiencies, as observed for the low FRET state (Fig. 4c–e). Cysteine residues from opposing subunits are considerably further apart (more than ~60 Å $C_\alpha$ to $C_\alpha$ and distance expected to be longer with fluorophores, Fig. 4a), which would yield FRET efficiencies below 30%. Very low FRET efficiency results in a small number of photons in the acceptor emission

channel making it difficult to detect FRET in these molecules. This could explain why FRET corresponding to this distance could not be observed in FRET histograms. When labeled with a donor-acceptor pair, Cys387 in adjacent subunits should thus act as a sensitive probe for conformational changes in SthK.

First, we monitored the FRET efficiency distributions of WT SthK in the absence and presence of cAMP. According to Step Transition and State Identification (STaSI) analysis[42], apo SthK displays two distinct FRET levels (Supplementary Fig. 9, FRET efficiencies ~0.92 and ~0.81) indicative of two conformations, which we tentatively assigned to closed states of high and intermediate FRET (Fig. 4b). In the presence of cAMP, we observed additional transitions to a third level of considerably lower FRET efficiency (~0.58) and of low abundance, consistent with a low-populated open state (Figs. 1f and 4c and Supplementary Fig. 9).

In the absence of cAMP, SthK Y26F and R120A led to the same high and intermediate FRET levels as WT SthK with similar state distributions (Supplementary Fig. 8b). In the presence of cAMP, both mutants also feature the lower FRET efficiency state, indicative of an open conformation. The occupancy of the open states in the mutants is higher than that of WT SthK, which is consistent with their increased Po (Fig. 4c–e). In addition, the occupancy of the intermediate FRET state is also enriched in the mutants compared to WT SthK, with the highest proportion observed for SthK R120A, possibly explaining why we could capture multiple intermediate states with cryo-EM with this mutant (Fig. 4b–e and Supplementary Fig. 8b, c). In summary, our sm-FRET experiments confirm the existence of gating intermediates identified in our structural studies, are consistent with the increased activity in the mutant channels, and importantly, suggest that WT SthK, SthK Y26F, and SthK R120A sample similar conformations during gating.

### Gating intermediates correlate with the open state lifetimes

Mutations in the VSD of SthK increased the open probability of the channel (Fig. 1f, g), which allowed us to structurally resolve multiple gating intermediates. However, despite similar Po values at 0 mV, SthK Y26F and SthK R120A displayed different intermediates in our structural analysis. To gain insight into how the different structural conformations along the gating pathway were stabilized, we analyzed the single-channel kinetics from recordings as in Fig. 1.

The open dwell-time distributions were fitted well with one exponential component for all three channels tested from −100 to +100 mV (distributions for representative bilayer recordings are shown in Supplementary Fig. 8f, g). The closed dwell time distributions generally are heterogenous, contain several exponential components, ranging from seconds to microseconds, and have too few events per time bin for accurate fitting and comparisons, and we did not analyze them further. The mean open dwell times for WT SthK are briefest over the entire voltage range, while SthK R120A has longest open dwell times (Fig. 4f).

The low Po (at 0 mV) due to the short-lived high-energy open states might explain the absence of a structurally resolved open state in WT SthK[28]. In SthK Y26F the conformational landscape is altered and while the channel features longer openings than WT SthK, the open state is likely still not sufficiently stabilized to be captured on cryo-EM grids, which resulted in an activated, pre-open intermediate. For SthK R120A the open states are sufficiently stabilized relative to WT SthK and SthK Y26F on cryo-EM grids, possibly because of the further increased mean open dwell times (Fig. 4f). Overall, although the single-channel characteristics at 0 mV are not drastically different between the three channels studied here, the differences in the durations of the mean open times become clearer at more depolarized voltages. This highlights the increased coupling between cAMP binding and channel opening in the gain-of-function mutants with enhanced voltage sensor mobility which may explain the observed structural differences (Fig. 4f and Supplementary Fig. 7).

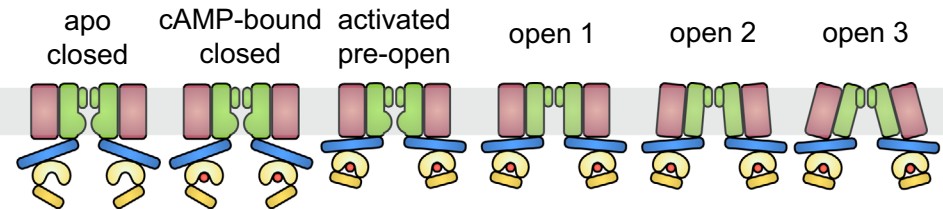

**Fig. 5 | CNG channel on-pathway gating intermediates.** Schematic model of CNG channel gating from an apo closed state, to the fully open state including intermediate states identified in this study. Channels are represented as VSD (red), pore domain (green), C-linker (blue), CNBD (yellow), and cAMP (orange). Membrane is indicated in gray.

## Discussion

Previous efforts to structurally determine functional states along the ligand-activation pathway for cyclic nucleotide-modulated channels (CNG and HCN channels) have so far been limited to only end states[14–16,28,33,43,44]. Here, we were able to stabilize several on-pathway gating intermediates and obtained cryo-EM structures of these otherwise high-energy, meta-stable conformations. The structures of such intermediates are crucial to understanding the function and regulation of ion channels, and a prerequisite for the design of drugs specifically targeting such intermediates to treat channelopathies.

To stabilize intermediates, we modified the conformational landscape via gain-of-function mutations in the voltage sensor domain of SthK. These mutations were subtle and did not change the structures of the closed states as compared to WT SthK. Even locally, around the sites of mutation, the structures were identical to WT SthK. To our delight, the mutations stabilized the channels in conformations that displayed activated ligand-binding domains, and pores with different degrees of opening at the gates. The C-linker/CNBD of the activated, pre-open state (SthK Y26F) is identical to those in the three open states (SthK R120A). Furthermore, the conformational changes between the CNBD/C-linker domains in the closed and the activated/open state observed in our structures, are consistent with AFM and DEER measurements of SthK as well as with structures of the closed and open states of SthK and of the eukaryotic CNG channels TAX-4 and CNGA1[14,16,23,27,34]. The fact that mutations in the VSD lead to conformational changes in the CNBDs and the pore, highlights the allosteric link between the voltage sensor domain, ligand-binding domain, and the channel pore (Fig. 5)[17], captured here with cryo-EM.

The gain-of-function mutations were designed based on the apo WT SthK structure, with the assumption that the S4 helix in the VSD, which is stabilized at rest by a network of interactions with adjacent transmembrane helices, inhibits the ligand-induced movements required for channel opening. Disrupting those interactions was intended to allow repositioning of the S4 helix and thus to activate the channel more easily. Indeed, most of our mutants showed either gain-of-function or no change, functionally. The structures of the two mutants investigated here, showed that their closed state structures were unchanged from closed WT SthK (apo and cAMP-bound)[28]. The intermediate and open structures indicated that for the S6 expansion to open the pore, S5 needs to move outwards, which is sterically impeded by the resting position of the S4 helix. Indeed, our open structures display an outward kink at the intracellular end of S4, which was sufficient to allow the S6 expansion and opening of the pore. Based on our findings, we propose that the voltage sensor domain, specifically, S4, plays an inhibitory role on SthK channel gating, and that this inhibition can be relieved with membrane depolarization. Membrane depolarization would lead to a similar small, lateral VSD repositioning and release of inhibition as seen in the open mutant structures by allowing the conformational changes in the C-linker/CNBD upon ligand binding and S6 expansion to occur (Fig. 5).

This small and rather unexpected voltage sensor movement relative to what has been proposed for voltage-gated Kv or HCN channels might reflect the relatively low gating charge of SthK[24,43,45–47]. In HCN channels, which are activated by hyperpolarization instead of depolarization like SthK and CNG channels, and have non domain-swapped VSDs like SthK and CNG channels, S4 is considerably longer than in other voltage-gated channels[48]. The S4 helix was shown to move downwards into the cytosol by two helical turns upon hyperpolarization in HCN channels. To accommodate this movement, S4 undergoes a large kink at the intracellular membrane boundary positioning the lower part of S4 parallel to the membrane[43,46,47]. In contrast, BK channels, that are only weakly voltage dependent (z ~0.6–1.5, compared to ~0.8 in SthK, and ~0.2 in CNG channels[24,49,50]), are proposed to undergo only minimal conformational changes at the intracellular end of S4[50], similar to the changes observed here. Furthermore, activity of eukaryotic CNG channels, with an even lower voltage-dependence than SthK, are also modulated by voltage such that their open probability and mean open time are slightly increased with depolarization, suggesting that a similar inhibitory VSD mechanism, as proposed here for SthK, may be at work in these channels[29,51].

Ligand binding is the crucial energy input for conformational changes in CNG channels. We propose that the partial agonist nature of cAMP for SthK allowed us to capture on-pathway gating intermediates rather than just end states. Full agonists, like cGMP for CNGA1 channels, are very effective in stabilizing open states, which may explain why only end open states have been captured using structural methods[16]. Conversely, partial agonists are less effective in stabilizing open states, which can result in relative stabilization of intermediate states. Our work introduces a novel way to determine structures of high energy states by employing partial agonists in combination with mutations that shift agonist efficacy.

In SthK, cAMP is a partial agonist at 0 mV. However, depolarization can increase cAMP efficacy, and gain-of-function mutations in the voltage sensor can lead to an almost unitary single-channel Po, suggesting that the partial agonist status of cAMP can be modulated by voltage. This i) shows directly that the voltage sensing mechanism and cAMP-gating are allosterically linked, and ii) suggests that cAMP may be a full agonist if not for the inhibitory action of the VSD, pointing to an intriguing parallel to eukaryotic CNG channels. One could speculate that during evolutionary integration of eukaryotic CNG channels into more complex signaling pathways (as compared to bacteria), the VSD may have lost most of its inhibitory action making cyclic nucleotides powerful agonists of CNG channels with only vestigial voltage-dependence[1].

In summary, we showed here structural evidence of allostery in CNG channels, by introducing perturbations in the voltage-sensor domain and capturing conformational changes taking place along the ligand-activation pathway. We took advantage of the dual activation of SthK by voltage and cAMP and the partial agonist nature of cAMP to capture on-pathway gating intermediates, which are normally too unstable to be captured with cryo-EM. Based on our data, we defined a detailed ligand-activation pathway for SthK and proposed an inhibitory role for the voltage-sensor domain (Fig. 5). Our findings shed

direct light on key conformational transitions and can be used to understand similar pathways in eukaryotic CNG channels.

## Methods

### Molecular biology

The gene for WT SthK was cloned into the pCGFP-BC vector[24,52]. Site-directed mutagenesis was performed using a site-directed mutagenesis kit (Agilent), plasmid preparations with the QIAprep Spin Miniprep Kit (Qiagen). Primers (Supplementary Table 5) were ordered from Integrated DNA Technologies (IDT). The plasmid for membrane scaffold protein 1E3 (MSP1E3) was purchased from Addgene (#20064) and amplified in *E. coli* XL-1 blue cells (Agilent). DNA sequencing was performed at Genewiz Inc.

### Expression and purification of SthK and MSP1E3

All SthK constructs were expressed in *E. coli* C41 (DE3) cells (Lucigen). Cells were grown in LB media at 37 °C in the presence of 100 mg/L ampicillin. At an $OD_{600nm}$ of 0.6, cells were transferred to 20 °C for further growth until $OD_{600nm}$ reached 0.8 at which point protein expression was induced with 0.5 mM IPTG. The cells were harvested after 12 h by centrifugation at $7500 \times g$ for 10 min. The pellets were resuspended in 25 mL buffer containing 50 mM HEPES, 200 mM KCl, pH 8 under constant agitation at 4 °C for 30 min in the presence of 1 mg DNase (Sigma), 1 mg lysozyme (Sigma) and 8.5 mg PMSF. All further steps were performed at 4 °C unless otherwise specified. Subsequently, the volume was adjusted to 50 mL with distilled and deionized water. Leupeptin/Pepstatin (0.95/1.4 µg/mL), cAMP (200 µM), cOmplete ULTRA EDTA-free protease inhibitor (Roche) and additional PMSF (170 µg/mL) were added, and the cells were broken by sonication on ice (Sonic Desmembrator 500, Fisher Scientific). 30 mM n-Dodecyl-β-D-Maltopyranoside (β-DDM, Anatrace) was added to solubilize membranes for 2 h. Cell debris and insoluble components were pelleted by centrifugation ($37000 \times g$, 45 min). The supernatant was supplemented with 40 mM imidazole and was filtered through a 0.22 µm filter before loading onto a 5 mL HiTrap chelating HP column (GE Life Sciences) charged with $Co^{2+}$ and equilibrated with running buffer (20 mM HEPES, 100 mM KCl, 1 mM β-DDM, 200 µM cAMP, 40 mM imidazole, pH 7). The column was washed with 50 mL running buffer, and the protein was eluted with elution buffer (20 mM HEPES, 100 mM KCl, 1 mM β-DDM, 200 µM cAMP, 400 mM imidazole, pH 7). The protein was concentrated to ~10 mg/mL using a concentrator with a cut-off molecular weight of 100 kDa (Amicon Ultra, Millipore). SthK channels were further purified by gel filtration using a Superdex 200 10/300 GL column (GE Life Sciences) at room temperature equilibrated with buffer containing 20 mM HEPES, 100 mM KCl, 1 mM β-DDM and 200 µM cAMP, pH 7.

MSP1E3 was expressed in *E. coli* BL21 (DE3) (New England Biolabs) in LB media at 37 °C (50 µg/mL kanamycin). Protein expression was induced with 1 mM IPTG for 4 h at an $OD_{600nm}$ of 0.8. All following steps were carried out at 4 °C. Cells were harvested at $7500 \times g$ for 10 min and resuspended in 50 mL resuspension buffer containing 40 mM Tris/HCl (pH 8 at RT), 300 mM NaCl. Leupeptin/Pepstatin (0.95/1.4 µg/mL), PMSF (170 µg/mL), 1 mg DNase, 1 mg lysozyme, cOmplete ULTRA EDTA-free protease inhibitor (Roche) and 1% Triton X-100 were added, and the cells were broken by sonication on ice (Sonic Desmembrator 500, Fisher Scientific). The suspension was cleared by centrifugation at $37000 \times g$ for 1 h, the supernatant was filtered through a 0.22 µm filter and loaded onto a HiTrap chelating HP $Ni^{2+}$ column equilibrated with loading buffer (40 mM Tris/HCl, pH 8 at RT), 300 mM NaCl, 1% Triton X-100). The column was washed with 50 mL of each of the following buffers: buffer I (40 mM Tris/HCl (pH 8 at RT), 300 mM NaCl, 1% Triton X-100, 50 mM sodium cholate, 20 mM imidazole), buffer II (40 mM Tris/HCl (pH 8 at RT), 300 mM NaCl, 1% Triton X-100, 50 mM imidazole). Protein was eluted with buffer

containing 40 mM Tris/HCl (pH 8 at RT), 300 mM NaCl, 1% Triton X-100 and 400 mM imidazole. The eluted protein was concentrated with a cut-off molecular weight of 30 kDa concentrator (Amicon Ultra, Millipore) and applied to a PD-10 column (GE Lifesciences) equilibrated in buffer containing 50 mM Tris/HCl (pH 8 at RT), 150 mM KCl, 0.5 mM EDTA for desalting. The collected protein flow-through was concentrated to ~10 mg/mL, flash frozen in liquid nitrogen, and stored at −80 °C for future use.

### Proteoliposome and nanodisc reconstitutions

To form SthK-containing liposomes, 1,2-dioleogyl-sn-glycero-3-phosphochline (DOPC), 1-palmitoyl-2-oleoyl-sn-glycero-3-phospho-(1′-rac-glycerol) (POPG) and 1′,3′-bis[1,2-dioleoyl-sn-glycero-3-phospho]-glycerol (18:1 Cardiolipin) in chloroform were mixed at a ratio of 5:3:2 (w/w/w) and dried under continuous $N_2$ flow in a glass tube. The lipids were washed in *n*-pentane and dried again under $N_2$ flow. Lipids were solubilized in buffer containing 10 mM HEPES, 400 mM KCl, 5 mM NMDG, 34 mM CHAPS, pH 7.6. SthK channels were added to the solubilized lipids at a ratio of 30 µg SthK/mg lipids for single channel recordings. The mixture was run through a 18 mL column packed with Sephadex G50 fine beads (GE Life Sciences) to remove the detergent. Proteoliposomes were collected, aliquoted, flash frozen in liquid nitrogen, and stored at −80 °C for future electrophysiology studies.

The same lipids composition was used to form nanodiscs for cryo-EM studies. Before nanodisc assembly, SthK protein was concentrated to ~10 mg/mL. The protein, MSP1E3 and lipids were mixed at a molar ratio of 1 (SthK monomer):1.5:50. 1 mM cAMP was added and the mixture was incubated for 1 h. Bio-beads (SM2, Bio-Rad) equilibrated in buffer containing 20 mM HEPES, 100 mM KCl, 200 µM cAMP, pH 7 were added (20 mg Bio-beads/100 µL sample) to initiate detergent removal while gently shaking. After 2 h, the sample was transferred to tubes containing fresh Bio-beads and further incubated overnight while gently shaking. Bio-beads were removed, and the sample was subjected to centrifugation for 10 min to remove aggregates before filtering the sample through a 0.22 µm SpinX filter (Costar). The sample was loaded onto a Superose 6 Increase column (GE Life Sciences) equilibrated in running buffer (20 mM HEPES, 100 mM KCl, 200 µM cAMP, pH 7). SthK containing nanodiscs were concentrated to 7–8 mg/mL with a cut-off molecular weight of 100 kDa concentrator (Amicon Ultra. Millipore). Both the proteoliposome and the nanodisc reconstitutions were carried out at room temperature.

Nanodiscs for sm-FRET were formed in a similar way. SthK channels, MSP1E3, and POPG were mixed at a molar ratio of 1 (SthK monomer):1:70. The two Bio-bead steps were performed at 4 °C. Additional purification by gel filtration was performed as for the cryo-EM sample preparation, the collected SthK-containing nanodiscs were not concentrated for sm-FRET experiments.

### Electrophysiology

Single channel recordings were performed in a horizontal lipid bilayer system where DPhPC lipid (6.25 mg/mL) dissolved in *n*-decane was used for bilayer painting over a hole (~200 µm diameter) in a plastic partition that separates the bottom and top chambers. Both chambers were filled with buffer containing 10 mM HEPES, 97 mM KCl, pH adjusted to 7 using 3 mM KOH. The bottom chamber additionally contained 200 µM cAMP to exclusively activate the channels facing the bottom chamber with their CNBDs. During the experiments, upon bilayer formation, the freshly thawed proteoliposomes were applied to the top chamber. All signals were amplified by a Axopatch 200 A (Molecular Devices), filtered at 1 kHz with an 8-pole low-pass Bessel filter and digitized at 20 kHz using a Digidata 1440 A digitizer (Molecular Devices) controlled by Clampex 10 (Molecular Devices). Single channel data analysis was performed in Clampfit 10 (Molecular Devices). The voltage dependence of SthK was fitted with a modified

Boltzmann Equation (Eq. 1).

$$Po(V) = \frac{Po_{max} - Po_{min}}{1 + e^{\frac{-F \cdot z \cdot (V - V_{half})}{R \cdot T}}} + Po_{min} \qquad (1)$$

where $F$ is the Faraday constant, $z$ the gating charge, $V_{half}$ the voltage of half activation, $R$ the universal gas constant, $T$ the temperature. $Po_{max}$ and $Po_{min}$ are maximum and minimum open probability, respectively. All data points were averaged from at least three recordings. The number of recordings for each experiment $n$ is given in the figure legends and in Supplementary Table 1.

The open dwell times were obtained by fitting the all open-events histogram with a single component exponential log probability function in Clampfit 10. The $EC_{50}$ values for all constructs were obtained by fitting the Po as function of the cAMP concentration using a Hill equation (Eq. 2).

$$Po([cAMP]) = Po_{min} + \frac{Po_{max} - Po_{min}}{1 + \left(\frac{EC_{50}}{[cAMP]}\right)^H} \qquad (2)$$

$EC_{50}$ is the cAMP concentration of half activation, $H$ is the Hill coefficient.

## Cryo-EM grid preparation and data collection

UltrAu-Foil R1.2/1.3 300-mesh gold grids (Quantifoil) were glow-discharged for 80 s. SthK nanodisc samples (7–8 mg/mL) were supplemented with 2 mM cAMP to saturate all possible binding sites, and 3 mM fluorinated Fos-choline 8 (Anatrace) for better particle distribution on the grids. Plunge freezing was carried out with a Vitrobot Mark IV (FEI). 3.5 µL of sample were incubated on the grid for 10 s at 100% humidity and 22 °C. The grid was blotted for 1.5 s with 0 blot force, and instantly plunge frozen in liquid ethane.

For SthK Y26F, SerialEM was used for data collection on a 300 kV Titan Krios (FEI) equipped with a K2 summit direct electron detector (Gatan) in super resolution mode. The calibrated pixel size was 1.06 Å/pixel and the nominal defocus was set between −1 and −3 µm. Movies containing 40 frames over 8 s were collected with an accumulated dose of 54.6 e$^-$/Å$^2$ (1.365 e$^-$/Å$^2$/frame). For SthK R120A, Leginon[53] was used for data collection on a 300 kV Titan Krios equipped with a K3 summit camera in super resolution mode. The pixel size was 1.09 Å/pixel and the defocus values were set between −1.3 and −2 µm. Movies containing 42 frames over 4.2 s were collected with an accumulated dose of 61.28 e$^-$/Å$^2$ (1.459 e$^-$/Å$^2$/frame).

## Cryo-EM data analysis

The dataset for SthK Y26F was analyzed in Relion 3.0_beta. Motion correction was done using Relion's own implementation, and CTF estimation without dose-weighting was done with CTFFIND 4.1[54] for 6748 micrographs. A total of 1,884,073 two-times binned particles were initially extracted and subjected to three rounds of 2D classification to remove junk particles. 1,324,724 particles were selected and re-extracted without binning. Detailed processing is shown in Supplementary Fig. 2. Eventually, 109,647 particles from 4 pre-open, activated classes (each from a different 3D classification along data analysis) were pooled and refined into one SthK Y26F pre-open state map with C4 symmetry. All identified closed state particles from each round of 3D classification, totaling 1,112,621, were combined and another round of 3D classification was performed, from which the SthK Y26F closed state map was finally constructed with 46,786 particles. All 3D refinements were done in two rounds where the second round used a solvent mask for more focused refinement on the protein. Two rounds of CTF refinement and Bayesian polishing were performed to improve the resolution and quality of the final maps. The resolution of

the final map reached 3.8 Å for pre-open SthK Y26F and 3 Å for closed SthK Y26F (FSC$_{0.143}$).

The dataset for SthK R120A was analyzed with Relion 3.1. Motion correction and CTF estimation were done similarly to that of SthK Y26F with 8284 images. Initially, 6,352,127 binned particles were extracted for five rounds of 2D classification, after which 3,288,140 particles were selected and re-extracted without binning. Another two rounds of 2D classification were performed to further remove bad particles, resulting in 3,004,702 particles that were then subjected to 3D classification. Detailed processing is shown in Supplementary Fig. 4. The final maps for SthK R120A closed state (799349 particles), open state 1 (O1, 50037 particles), O2 state (149792 particles) and O3 state (46629 particles) were subjected to two rounds of CTF refinement and Bayesian polishing, which resulted in resolutions of 2.9, 4.2, 3.6, and 3.6 Å with C4 symmetry, respectively (FSC$_{0.143}$).

## Model building and validation

All models were refined in Phenix (version 1.18.2) and manually adjusted in Coot. The models consist of the following amino acid residues: closed SthK Y26F 10–65, 74–418; pre-open SthK Y26F 10–62, 76–420; closed SthK R120A 10–64, 76–415; SthK R120A 10–62, 81–125 and 132–417, SthK R120A O2 and O3 10–62, 81–125 and 130–420. All maps show some density for lipids at the periphery of the protein. Since theses densities in most places are not strong enough to model the entire lipid, most of these additional densities were modeled with only the alkyl chains. Model validation was performed in MolProbity. Pore size measurements were done using the Hole program implemented in Coot (Smart et al., 1996).

## Labeling of SthK in nanodiscs for single molecule FRET

The nanodiscs used for sm-FRET were stored at 4 °C. For each day of data acquisition, a fraction of the nanodiscs was freshly labeled with fluorophores. Stock solutions of 1 mM maleimide-linked Alexa 555 (donor) and Alexa 647 (acceptor) fluorophores (Invitrogen) in DMSO were pre-mixed in a 1:4 (donor:acceptor) molar ratio in a separate tube containing 100 µL of buffer A (20 mM Hepes, 0.1 M KCl, pH 7.4). The pre-mixed fluorophores were then added to 25–50 µg of SthK-containing nanodiscs in buffer A to achieve a final volume of 500 µL and final fluorophore concentrations of 600 nM Alexa 555 and 2.4 µM Alexa 647 (final DMSO < 0.3% in the labeling reaction). The sample was then rotated at room temperature for 30 min while being protected from ambient light to allow the fluorophores to label the SthK protein. To decrease the concentration of free fluorophores in the solution during the binding of the SthK molecules to the slide, the sample was diluted 10-fold with buffer A before being applied to the sm-FRET slide. The final buffer composition used during sm-FRET measurements is described in the Slide preparation section.

## Slide preparation for sm-FRET

Experimental methods concerning sm-FRET slide preparation, data collection, and data analysis are as previously described (Durham et al., 2020). In brief, glass coverslips were used to immobilize sample molecules for sm-FRET measurements. The coverslips were first cleaned via bath sonication in Liquinox phosphate-free detergent (Fisher Scientific) and then acetone. Further cleaning was achieved by incubating the slides in a 4.3% NH$_4$OH and 4.3% H$_2$O$_2$ solution at 70 °C followed by plasma cleaning in a Harrick Plasma PDC-32G Plasma Cleaner. After cleaning of the slide surface, the slide was treated to prepare a Streptavidin-coated surface for the attachment of SthK nanodiscs. This process was begun via aminosilanization of the slide surface with Vectabond reagent (Vector Laboratories). This step was followed by polyethylene-glycol (PEG) treatment of the slide surface with 0.25% w/w 5 kDa biotin-terminated PEG (NOF Corp., Tokyo, Japan) and 25% w/w 5 kDa mPEG succinimidyl carbonate (Laysan Bio Inc., Arab, AL). This initial PEG treatment was followed by a secondary PEG

treatment with 25 mM short-chain 333 Da MS(PEG)4 Methyl-PEG-NHS-Ester Reagent (Thermo Scientific). These treatments resulted in a biotin-coated surface on the slide. Next, a microfluidics chamber consisting of an input port, a sample chamber, and an output port was constructed on the slide (Litwin et al., 2019). To coat the biotinylated surface with streptavidin molecules, buffer A supplemented with 0.2 mg/mL Streptavidin was injected into the chamber. After 10 min, unbound streptavidin was washed away, and fluorophore-labeled SthK protein sample was applied to the slide. Before the sm-FRET data acquisition the sample was allowed to adhere to the slide (10–15 min) and in the next step the chamber was washed with imaging buffer consisting of buffer A supplemented with reactive oxygen species scavenging solution (ROXS) components: 3.3% w/w glucose, 3 units/mL pyranose oxidase, 0.001% w/w catalase, 1 mM ascorbic acid, and 1 mM methyl viologen (all compounds from Sigma-Aldrich). In the case of cAMP experiments, 1 mM cAMP was added to the ROXS solution.

### Sm-FRET data acquisition

After the preparation of immobilized, fluorophore-labeled SthK nanodiscs on the slide surface, sm-FRET data was acquired from individual SthK channels using a MicroTime 200 Fluorescence Lifetime Microscope from PicoQuant. A donor excitation laser (532 nm; LDH-D-TA-530; Picoquant, Berlin, Germany) and acceptor excitation laser (637 nm; LDH-D-C-640; Picoquant) were used with a Pulsed Interleaved Excitation (PIE) scheme to excite the fluorophores. Emitted photons were collected back through the objective lens (100×1.4 numerical aperture; Olympus). Emission filters for the donor (550 nm; FF01-582/64; AHF, Tübingen-Pfrondorf, Germany or Semrock, Rochester, NY) and acceptor (650 nm 2XH690/70; AHF) were used to select photons for each detection channel. Photons were then passed to two SPAD photodiodes (SPCM CD3516H, Excelitas technologies, Waltham, MA) to determine the fluorescence intensity for each fluorophore. The donor and acceptor fluorescence intensities over time were recorded for one SthK channel at a time and later analyzed as described below.

### Sm-FRET data analysis

Only molecules that exhibited a single photobleaching step in each of the donor and acceptor channels were selected for analysis. This ensures that only one donor and one acceptor fluorophore are attached to each SthK molecule. Additionally, molecules were screened to include only those that exhibited anti-correlation between the donor and acceptor fluorescence, which ensures that the fluorophores were engaging in FRET prior to photobleaching. Molecules that did not exhibit these characteristics were not included in the final analysis. The final number of molecules that were included in the analysis for each condition were: WT apo: $n = 53$; WT cAMP: $n = 58$; SthK Y26F apo: $n = 53$; SthK Y26F cAMP: $n = 79$; SthK R120A apo: $n = 63$; SthK R120A cAMP: $n = 73$. The fluorescence intensity traces for the included molecules were corrected for differences in fluorophore quantum yield and detector efficiency. This correction was performed using the γ factor, which was calculated based on the ratio of change of the acceptor intensity to the change of the donor intensity upon acceptor photobleaching. The γ factor was then used to adjust one fluorophore's trace such that differences in quantum yield and detector efficiency were nullified[55]. The corrected donor and acceptor intensities over time were then used to calculate a FRET efficiency trace for each molecule. These traces were pooled for each condition and used to create the FRET efficiency distribution histograms for each condition. Using the pooled traces, Step Transition and State Identification (STaSI) analysis was performed using Matlab to determine the number of conformational states in each condition[42]. The smallest number of states that accurately describes the data as determined by the STaSI analysis was used as the final number of states for each condition. Using the results of the STaSI analysis and Origin software

(OriginLab), the FRET efficiency histograms for each condition were fit with 2–3 Gaussian curves to show the conformational states that make up the overall distributions. To create the contour maps that are associated with each FRET distribution histogram, the FRET efficiency traces for each condition were concatenated into a single trace containing the FRET efficiency data for all molecules of a given condition. This concatenated efficiency trace was subjected to a wavelet denoising process using Origin (OriginLab). The Haar wavelet type with periodic extension mode was used with a thresholding level of 2 to denoise the data. Level 2 denoising was selected as it was the lowest level of denoising that eliminated a majority of the noise component from the signal. The resulting denoised, concatenated trace was then divided into one-second-long segments. For each condition, these segments were traced on top of each other to create a heat map showing the overall distribution of FRET efficiencies. The percentage of a given FRET efficiency out of the total for that time point is shown according to the color scale.

### Figure preparation

All figure panels are prepared in Clampfit (Molecular Devices), Igor Pro (Wavemetrics), Chimera (UCSF)[56], or ChimeraX (UCSF)[57] before integration in Illustrator (Adobe).

### Reporting summary

Further information on research design is available in the Nature Portfolio Reporting Summary linked to this article.

### Data availability

Data supporting the findings of this manuscript are available from the corresponding author upon request. A reporting summary for this article is available as a Supplementary Information file. The maps for SthK in different states have been deposited in the Electron Microscopy Data Bank (EMDB) under accession codes EMD-24670 (SthK Y26F closed state), EMD-24682 (SthK Y26F activated state), EMD-24681 (SthK R120A closed state), EMD-24692 (SthK R120A open state 1), EMD-24747 (SthK R120A open state 2), EMD-24746 (SthK R120A open state 3). Atomic coordinates for all structures have been deposited in the Protein Data Bank (PDB) with accession codes 7RSH (SthK Y26F closed state), 7RTJ (SthK Y26F activated state), 7RTF (SthK R120A closed state), 7RU0 (SthK R120A open state 1), 7RYS (SthK R120A open state 2), 7RYR (SthK R120A open state 3). The model for WT SthK (6CJU) and the corresponding cryo-EM density (EMD-7484) were used for comparisons. The source data underlying Figs. 1E, F, G; 4F; and Supplementary Figs. 1B, 3, 5, and 8A are available as a Source Data file. Source data are provided with this paper.

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

## Acknowledgements

We thank Lee Cohen-Gould and Devrim Acehan for help with sample preparation at Weill Cornell Medicine, Laura Yen and Daija Bobe for grid screening at the Simons Electron Microscopy Center and National Resource for Automated Molecular Microscopy located at the New York Structural Biology Center, which is supported by grants from the Simons Foundation (349247), NYSTAR, and the NIH National Institute of General Medical Sciences (GM103310), Chen Xu and Kangkang Song for data collection at UMass Medical School, William Rice and Bing Wang for data collection at NYU Langone Health's Cryo–Electron Microscopy Laboratory (RRID: SCR_019202). The work presented here was sponsored in part by the NIH (GM122528) to V.J. and (GM124451) to C.N.

## Author contributions

X.G. performed mutant screening, electrophysiology recordings and prepared cryo-EM samples, X.G. and C.F. prepared samples for sm-FRET and collected cryo-EM data. X.G. analyzed electrophysiology data. X.G., P.S., and C.F. analyzed cryo-EM data. V.B., R.D., and V.J. performed and analyzed sm-FRET experiments. X.G., P.S., and C.N. assembled the manuscript and wrote the paper with input from all authors.

## Competing interests

The authors declare no competing interests.
