## [Peer Review File · Nature Communications]

Gating intermediates reveal inhibitory role of the voltage sensor in a cyclic nucleotide-modulated ion channelReviewers' Comments:

Reviewer #1:

Remarks to the Author:

This manuscript from the Nimigean lab, in collaboration with the Jayaraman lab, elucidated the structural mechanism underlying the activation of a bacterial cyclic nucleotide-modulated ion channel Sthk. Previous work from the same group has obtained a closed structure and a putative low-resolution open structure of the Sthk. This work, by using mutations in the voltage-sensing domain, revealed several high-resolution open cryo-EM structures, which suggested that the channel had intermediate states throughout its conformational rearrangements needed for opening the pore. Sthk is a good model channel in understanding ligand-gated channels and other allosteric macromolecules. Considering its stronger voltage dependence, Sthk behaves more like a HCN channel but with a reversed voltage-dependent gating polarity. Cyclic nucleotides alone can barely open the channel, which is similar to HCN channels. With a z value of 0.8 calculated for the G-V relationship, the wild-type Sthk is also more voltage dependent than classic CNG channels. These cryo-EM structures of Sthk provided interesting and important advances in understanding the CNBD channel family particularly. The design of the single-molecule FRET was reasonable and can potentially support the idea of multiple intermediate states suggested by the cryo-EM structures. While the cryo-EM part was nicely done, some revisions and clarifications in the smFRET analysis are needed before publication.

Main issues:

1. The labeling of one cysteine per subunit (two fluorophores per tetramer) made the donor/acceptor positions unfixed. With a $R_0 = 51 \text{ \AA}$ for the FRET pair, FRET between both adjacent and diagonal subunits could be noticeable. On the Page 11, line 258, please clarify the rationale underlying the statement that distances longer than 55 \AA cannot be reliably measured with a R_0 of 51 \AA ?
2. Page 13: "Apo SthK displays two distinct FRET levels (FRET efficiencies ~ 0.92 and ~ 0.81) indicative of two conformations". Some explanation probably is needed to convincingly conclude that there are two distinct FRET levels in the contour histogram of the FRET efficiency distribution. Looks like the binned counting doesn't have enough information to constrain the fitting. What are the criteria for the gaussian fitting to determine FRET levels? In addition, how could the FRET efficiency be more than 1 in Fig. 4, B-E?
3. For the purpose of identifying intermediate states, why not using the Hidden Markov Model or something similar to analyze the smFRET traces, which would give the best idea of what the transition rates, states and paths are for the scheme? The binned counts in contour histogram discarded the temporal information.
4. From the table S1, the z value of the Y26F and the R120A decreased to mean values of 0.6 and 0.7 respectively, but apparently as stated in the paper, no significant difference comparing to the value of 0.8 for the wild type. Looks like the difference of the mutations is most obvious in the voltage-independent openings. Using a simple G-V Boltzmann fitting as used in this paper, a Shaker potassium channel would not generate 12-13 gating charges per channel. The 12-13 gating charges were calculated using the limiting slope analysis by measuring the very low P_o at very hyperpolarized voltages. If using the limiting slope analysis, the difference might be revealed between the mutants and the wild-type Sthk. Although I understand if time will not be allowed for additional experiments, the description on Page 4, line 85 needs a modification since the numbers of charges cannot be compared between these two cases. Also the HCN channels have less charge movement measured compared to the Shaker.
5. In the Fig. 1G, the Y26F and R120A showed a decreased Hill coefficient, suggesting some potential change in the cooperativity of the channel opening. Any interpretations on this change?

6. In the method section "the fluorescence intensity traces for the included molecules were subjected to various corrections for factors such as background noise, cross-talk between fluorophore channels, direct acceptor excitation, differences in quantum yield between fluorophores, and differences in detector efficiency". More details and the parameters of correction factors will be needed here for readers. In addition, why to choose the threshold level 2 to denoise the data?

7. Since the Sthk shares some interesting similarities with HCN channels, so comparisons of the Sthk and HCN voltage sensing might worth mentioning in the paper. Perhaps also cite papers on HCN voltage sensing mechanisms in the discussion part (page 16). The inhibitory role of the S4 helix for the HCN activation has been proposed and the kink of the S4 helix has also been observed in HCN previously.

8. Fig S7, B and C, I would not to call it "conformational landscape", since it specifically means a plot of free energy versus different conformational states in the field of thermodynamics of proteins.

9. Perhaps not to introduce CNG channels as "modulated by voltage" in the Page 3, line 57. In general, CNG channels are considered voltage insensitive but with slight outward rectification at positive voltages, which remains a mystery about its mechanism.

Reviewer #2:

Remarks to the Author:

Gao et al. use gain of function mutants in conjunction with a partial agonist to observe multiple conformations of cAMP-bound SthK cyclic nucleotide-gated channels in closed, apparent pre-active and open conformations. The observed structures suggest that small motions at the bottom of S4 and S5 helices promote activation of C-Linkers and cAMP-bound CNBDs and make way for opening of S6 and pore that may represent important gating intermediates between closed and fully open conformations. Furthermore, they show that pre-active vs open conformations observed in the two mutants correlate with lateral compaction of the CNBDs as measured by smFRET as well as single channel open lifetimes. This excellent work provides a structural look at potential gating intermediates in a CNG channel, and will be of broad interest to both biophysicists and for drug design targeting CNG channels.

Comments:

1. Given the location of R120 near the bottom of the S4 helix, structural changes in this region as observed in the open conformations can be readily rationalized. For Y26F which is a few turns above this region, can the authors speculate on why this mutation also confers a change near the bottom of S4 without any obvious changes in the structure at the mutation site or its interaction partner R111 (at least as far as I could tell from the figures). Was there perhaps a change in density suggesting a more flexible S4?

2. The author's suggest that longer mean open times in R120A vs Y26F are why open conformations were only observed for R120A. However, very little of the raw open time distributions are shown, and it would be nice to see more of the data that is summarized in Fig. 4F (for example, expand Fig. 7F to include all three channels and perhaps also a voltage closer to 0 mV to better correspond with the structures). Also, in Fig. 4B-E it appears that compact conformations are also longer lived in R120A. Is this true?

Minor Comments:

1. Lines 293-294: I don't understand why there are not sufficient closed events for accurate closed duration distributions. The data in Figs. 1D and S7D suggest otherwise.
2. Lines 253-255: Contrary to what is stated here, Fig. 8A suggests that C153A reduces the Po of R120A in particular.
3. Line 210: Define "slight S4 movement".

Marcel Goldschen-Ohm

Reviewer #3:

Remarks to the Author:

This manuscript by Gao et al. aims to determine the structures of intermediate states during SthK channel opening. SthK is a prokaryotic CNG channel with properties and structures common with vertebrate CNG channels, which have important physiological functions and whose dysfunction causes severe visual disorders. Structures of closed and fully open CNG channels have been obtained, but numerous questions remain to be elucidated on the gating process. Taking advantage of a partial agonist and a modest voltage dependence of SthK gating, and by using two gain-of-function mutations (Y20F and R120A) in the VSD, the authors stabilized and obtained the cryo-EM structures of several intermediate gating states of SthK. The authors then used sm-FRET to verify the existence of multiple states and single-channel analysis to correlate the structures to functional behaviors. This work successfully captures, for the first time, various transition states of a CNG channel en route from close to open, providing new mechanistic insights into allosteric gating of CNG channels and highlighting an inhibitory role of the VSD in the gating process. The methods used in this study could also be used to investigate other channels with similar gating mechanisms.

The experiments are well designed and executed, the data are convincing, the results are clear and interesting, and the paper is well written.

However, an important piece of information is missing: the structures of the apo state of both Y26F and R120A mutant channels. It is possible, or even highly likely, that these structures are identical to that of apo WT SthK. Still, it would be valuable to show that this is indeed the case. There is, however, a possibility that some mutant channels may display some global or local conformational changes (e.g., in the VSD, S4-S5 linker, S5 or S6) that are caused by the mutations and that can be captured by cryo-EM. It would be interesting, and indeed, important to examine this possibility. In this context, it is useful to note that a single amino acid mutation in the *C. elegans* CNG channel TAX-4 causes the channel to open and produces an open-state structure in the absence of cyclic nucleotides (Zheng et al., *Communications Biology*, 2022).

Minor comments/questions:

1. It would be helpful to zoom in around F26 and A120 and show the density maps and model fit to see the loss of interactions engaged by these residues and the local conformational changes caused by the mutations.
2. A displacement of the S4-S5 linker appears to be crucial for the pre-open state of Y26F. It would be helpful to show the interactions and Y26F mutation-induced changes, if there are any, between the S4-S5 linker and the C-linker.
3. In eukaryotic CNG channels, including TAX-4, CNGB1, CNGB1/CNGB1 and CNGB3/CNGB3, two hydrophobic residues form a gate in the central cavity, and this gate has been called a cavity gate or

central gate. The authors call the gate in SthK the "intracellular bundle crossing gate". Is the gate in SthK formed solely by I215 in SthK? Although it may be a matter of semantics, it is notable that I215 is 2-3 alpha-helical turns up than the "inner" or "bundle" gates in many other channels. A brief discussion about the similarity/dissimilarity of the activation gate in eukaryotic CNG channels and SthK may be useful.

4. In this study, SthK nanodiscs were reconstituted in DOPC:POPG:Cardiolipin. In previous studies by the same group, only POPG was used. Any particular reason(s) for this change of lipid composition?

5. Some parameters in structure refinement and validation seem to have room for improvement, including the clashscore, Ramachandran outliers, and preferred Ramachandran ratio.

6. There is an error in the citation of Carrasquel-Ursulaez et al.

We thank the reviewers for their insightful and detailed comments that led to a well improved version of our manuscript. All issues are addressed below in our point-by-point responses.

Reviewer #1 (Remarks to the Author):

This manuscript from the Nimigean lab, in collaboration with the Jayaraman lab, elucidated the structural mechanism underlying the activation of a bacterial cyclic nucleotide-modulated ion channel Sthk. Previous work from the same group has obtained a closed structure and a putative low-resolution open structure of the Sthk. This work, by using mutations in the voltage-sensing domain, revealed several high-resolution open cryo-EM structures, which suggested that the channel had intermediate states throughout its conformational rearrangements needed for opening the pore.

Sthk is a good model channel in understanding ligand-gated channels and other allosteric macromolecules. Considering its stronger voltage dependence, Sthk behaves more like a HCN channel but with a reversed voltage-dependent gating polarity. Cyclic nucleotides alone can barely open the channel, which is similar to HCN channels. With a z value of 0.8 calculated for the G-V relationship, the wild-type Sthk is also more voltage dependent than classic CNG channels. These cryo-EM structures of Sthk provided interesting and important advances in understanding the CNBD channel family particularly. The design of the single-molecule FRET was reasonable and can potentially support the idea of multiple intermediate states suggested by the cryo-EM structures. While the cryo-EM part was nicely done, some revisions and clarifications in the smFRET analysis are needed before publication.

Main issues:

1. The labeling of one cysteine per subunit (two fluorophores per tetramer) made the donor/acceptor positions unfixed. With a $R_0 = 51 \text{ \AA}$ for the FRET pair, FRET between both adjacent and diagonal subunits could be noticeable. On the Page 11, line 258, please clarify the rationale underlying the statement that distances longer than 55 \AA cannot be reliably measured with a R_0 of 51 \AA ?

The reviewer is correct that FRET can be observed for both adjacent and diagonally labeled SthK when using a FRET pair with $R_0 = 51 \text{ \AA}$. We have clarified in the relevant section (lines 278-291) that these two distances yield considerably different FRET efficiencies and that we did not observe molecules exhibiting the lower FRET efficiency that would be expected from the diagonally labeled molecules. We believe that this lack of low-FRET molecules is due to the fact that the low FRET efficiency results in a very small amount of FRET signal being contributed by the diagonally labeled molecules. In practice, these molecules appear similar to non-FRET molecules and are discarded from the analysis. Additionally, we have removed the line stating that an $R_0 = 51 \text{ \AA}$ is best suited for distances between 55 \AA and 27 \AA .

2. Page 13: "Apo SthK displays two distinct FRET levels (FRET efficiencies ~ 0.92 and ~ 0.81) indicative of two conformations". Some explanation probably is needed to convincingly conclude that there are two distinct FRET levels in the contour histogram of the FRET efficiency distribution. Looks like the binned counting doesn't have enough information to constrain the fitting. What are the criteria for the gaussian fitting to determine FRET levels? In addition, how could the FRET efficiency be more than 1 in Fig. 4, B-E?

Thank you for these comments, both are good points. In order to determine the number of conformational states in each condition, we used Step Transition and State Identification (STaSI) analysis (Shuang et al., 2014). This analysis identifies statistically independent states within the FRET efficiency trajectory and evaluates the number of states through the Minimum Description Length (MDL) function. For each possible number of states, the MDL function is used to calculate a score to identify the smallest number of states that accurately describes the data. The smaller the MDL value, the better the balance between simplicity and accuracy of the model. We have now included the charts of the MDL values for the four conditions shown in Figure 4 as Supplementary Fig. S9.

This analysis shows that the results for apo WT SthK are best described as two states. The three cAMP-conditions (WT SthK, Y26F, R120A), are well described by either three or four states, but three states represent the simplest model that accurately describes the data. Once the number of states for each condition is known, we use that information along with the FRET efficiency values of those identified states to fit Gaussian curves to the overall FRET efficiency histogram (Figure 4). This Gaussian fitting allows us to determine the error for the FRET efficiency values of the states and to determine the value and error for the percent occurrence of each state. We have modified the main text.

Lines 290-294: "First, we monitored the FRET efficiency distributions of WT SthK in the absence and presence of cAMP. According to Step Transition and State Identification (STaSI) analysis (Shuang et al., 2014), apo SthK displays two distinct FRET levels (Supplementary Fig. S9, FRET efficiencies ~ 0.92 and ~ 0.81) indicative of two conformations, which we tentatively assigned to closed states of high and intermediate FRET (Fig 4B)."

Additionally, we have extended the Methods section.

Lines 678-686: "The corrected donor and acceptor intensities over time were then used to calculate a FRET efficiency trace for each molecule. These traces were pooled for each condition and used to create the FRET efficiency distribution histograms for each condition. Using the pooled traces, Step Transition and State Identification (STaSI) analysis was performed to determine the number of conformational states in each condition (Shuang et al., 2014). The smallest number of states that accurately describes the data as determined by the STaSI analysis was used as the final number of states for each condition. Using the results of the STaSI analysis and Origin software (OriginLab), the FRET efficiency histograms for each condition were fit with 2-3 Gaussian curves to show the conformational states that make up the overall distributions."

The reason the FRET efficiency rises above a value of one in Figure 4 is due to experimental noise. To counteract this noise, a denoising process is conducted on the data which converts the data from the time domain into the wavelet domain, identifies and removes components of the data assumed to be due to noise, and converts the denoised data back into the time domain. This denoising almost completely eliminates the data points showing FRET efficiency higher than one. The denoising process was conducted before STaSI analysis was used to determine the number and value of the conformational states in each condition. In Figure 4, we show the data before the denoising process in the histograms to present the unaltered, raw data.

3. For the purpose of identifying intermediate states, why not using the Hidden Markov Model or something similar to analyze the smFRET traces, which would give the best idea of what the transition rates, states and paths are for the scheme? The binned counts in contour histogram discarded the temporal information.

In order to identify intermediates in our sm-FRET data we used Step Transition and State Identification analysis (STaSI), developed by the Landes Lab. We have now added this to the methods section in the revised version (lines 682-688). We did not use the sm-FRET traces for determining transition rates between states, as the sm-FRET data are collected in 5 ms bins, which reduces the time resolution considerably. Furthermore, transitions occur relatively infrequently (idealized FRET-efficiency traces in Figure 4B-E, Supplementary Table S4) providing us with too few events for detailed, kinetic analyses.

4. From the table S1, the z value of the Y26F and the R120A decreased to mean values of 0.6 and 0.7 respectively, but apparently as stated in the paper, no significant difference comparing to the value of 0.8 for the wild type. Looks like the difference of the mutations is most obvious in the voltage-independent openings. Using a simple G-V Boltzmann fitting as used in this paper, a Shaker potassium channel would not generate 12-13 gating charges per channel. The 12-13 gating charges were calculated using the limiting slope analysis by measuring the very low P_o at very hyperpolarized voltages. If using the limiting slope analysis, the difference might be revealed between the mutants and the wild-type Sthk. Although I understand if time will not be allowed for additional experiments, the description on Page 4, line 85 needs a modification since the numbers of charges cannot be compared between these two cases. Also the HCN channels have less charge movement measured compared to the Shaker.

Thank you for this detailed input. Indeed, we should have used z values that are more immediately comparable to the z value obtained in our analysis. We now changed the respective paragraph and compare SthK ($z \sim 0.8$) to Shaker ($z \sim 4$ (Zagotta et al., 1994)) and HCN ($z \sim 1$ (Ryu and Yellen, 2012)). The updated sentence is (lines 86-88): "... smaller gating charge (0.8 compared with 4 in Shaker and 1 in HCN, from Boltzmann fits of P_o vs. voltage plots) (Zagotta et al., 1994; Ryu and Yellen, 2012; Schmidpeter et al., 2018)."

5. In the Fig. 1G, the Y26F and R120A showed a decreased Hill coefficient, suggesting some potential change in the cooperativity of the channel opening. Any interpretations on this change?

The reviewer raises an interesting point. The changes in Hill coefficient indeed may reflect changes in cooperativity during cAMP-dependent channel activation. However, we do not have sufficient recordings to provide us with the necessary resolution to confidently draw conclusions about changes in cooperativity from our data. We added a sentence in the manuscript where we acknowledge the apparently different Hill coefficients and concede that more research is needed to determine whether cooperativity has been modified.

Lines 132-135: "The Hill coefficients required to fit the data in Fig. 1G are smaller for the mutants compared to WT (Supplementary Table S1). While this may indicate a change in cooperativity, more investigation is necessary on this topic."

6. In the method section "the fluorescence intensity traces for the included molecules were subjected to various corrections for factors such as background noise, cross-talk between fluorophore channels, direct acceptor excitation, differences in quantum yield between fluorophores, and differences in detector efficiency". More details and the parameters of correction factors will be needed here for readers. In addition, why to choose the threshold level 2 to denoise the data?

We have expanded the Methods section to give more information on the rationale behind the corrections to the sm-FRET data. The updated section is:

Lines 673-678: “The fluorescence intensity traces for the included molecules were corrected for differences in fluorophore quantum yield and detector efficiency. This correction was performed using the γ factor, which was calculated based on the ratio of change of the acceptor intensity to the change of the donor intensity upon acceptor photobleaching. The γ factor was then used to adjust one fluorophore’s trace such that differences in quantum yield and detector efficiency were nullified (Roy et al., 2008).”

When selecting a level of denoising for the sm-FRET data, we aimed to keep the data in its original, observed form as much as possible, while still eliminating the noise component. We used the sm-FRET signal that showed an efficiency value above one as an indicator since the signal above one is assumed to be due to experimental noise. We tested different levels of denoising and found that level two was the lowest level that eliminated a majority of the noise component. We therefore used level two denoising for our analysis.

We have updated the Methods section:

Lines 690-693: “The Haar wavelet type with periodic extension mode was used with a thresholding level of 2 to denoise the data. Level 2 denoising was selected as it was the lowest level of denoising that eliminated a majority of the noise component from the signal.”

7. Since the Sthk shares some interesting similarities with HCN channels, so comparisons of the Sthk and HCN voltage sensing might worth mentioning in the paper. Perhaps also cite papers on HCN voltage sensing mechanisms in the discussion part (page 16). The inhibitory role of the S4 helix for the HCN activation has been proposed and the kink of the S4 helix has also been observed in HCN previously.

Thank you for this comment. Indeed, SthK shares some features with CNG channels (e.g. activation by cAMP), and others with HCN channels (e.g. slow modulation by cAMP, lipid sensitivity), making it a model channel that can be used to study specific modalities of either of these channels. However, the voltage-dependent modulation of SthK is rather more similar to CNG than to HCN channels.

HCN channels are special with respect to their gating polarity as well as their S4 helix, which is believed to be the main determinant of voltage sensing. In HCN, S4 features eight positively charged residues and extends into the cytosol by two additional helical turns compared to other voltage-gated channels (Lee and MacKinnon, 2017). In contrast, SthK carries only four positive charges in S4 (R111, K114, R120, R124, Figure 1A), similar to CNG channels. SthK is also more similar to CNG channels in that its activity is increased by depolarizing potentials (although, as the reviewer also noted, with a very weak voltage dependence for CNG channels) and not hyperpolarizing potentials, like HCN.

Furthermore, the S4 in HCN channels was found to undergo a dramatic straight-to-bent transition as it changes from a resting to an activated conformation to relieve an inhibition of the C-linker-CNBD movement (Dai et al., 2019; Kasimova et al., 2019; Lee and MacKinnon, 2019). In contrast, our proposal in this manuscript is that the SthK S4 helix undergoes a small, lateral conformational change to relieve the steric inhibition placed on the iris-like pore dilation at the intracellular gate (S5-S6 helices), rather similar to an accordion movement (Figure 3D). These conformational changes are different from changes observed during hyperpolarization-dependent activation of HCN.

We clarified the text about the differences in voltage-dependence and by contrasting them with the HCN movements (we cited the relevant papers as well), see Discussion (lines 379-396).

8. Fig S7, B and C, I would not to call it “conformational landscape”, since it specifically means a plot of free energy versus different conformational states in the field of thermodynamics of proteins.

We changed conformational landscape to “Histograms of FRET efficiency distribution”.

9. Perhaps not to introduce CNG channels as "modulated by voltage" in the Page 3, line 57. In general, CNG channels are considered voltage insensitive but with slight outward rectification at positive voltages, which remains a mystery about its mechanism.

As the reviewer states, CNG channels show a “slight outward rectification at positive voltages”, which indicates that the activity is modulated by voltage. But point is taken, and we agree that it is very small, so we added a qualifier. Lines 56-57: “... only slightly modulated by voltage, although the mechanism is not yet understood”.

Reviewer #2 (Remarks to the Author):

Gao et al. use gain of function mutants in conjunction with a partial agonist to observe multiple conformations of cAMP-bound SthK cyclic nucleotide-gated channels in closed, apparent pre-active and open conformations. The observed structures suggest that small motions at the bottom of S4 and S5 helices promote activation of C-Linkers and cAMP-bound CNBDs and make way for opening of S6 and pore that may represent important gating intermediates between closed and fully open conformations. Furthermore, they show that pre-active vs open conformations observed in the two mutants correlate with lateral compaction of the CNBDs as measured by smFRET as well as single channel open lifetimes. This excellent work provides a structural look at potential gating intermediates in a CNG channel, and will be of broad interest to both biophysicists and for drug design targeting CNG channels.

Thank you for the positive assessment of our work.

Comments:

1. Given the location of R120 near the bottom of the S4 helix, structural changes in this region as observed in the open conformations can be readily rationalized. For Y26F which is a few turns above this region, can the authors speculate on why this mutation also confers a change near the bottom of S4 without any obvious changes in the structure at the mutation site or its interaction partner R111 (at least as far as I could tell from the figures). Was there perhaps a change in density suggesting a more flexible S4?

Indeed, there is no local change around the site of the Y26F mutation, or R120A mutation in the closed state (see new Supplementary Fig. S6). The idea behind both mutations was to disrupt interactions between S4 and adjacent helices (S1, S2 or S3), which would allow S4 to move easier and would destabilize the closed relative to the open state. For SthK Y26F for example, S4 is more loosely positioned, since one of the four basic residues (Arg111) no longer engages in interactions with surrounding helices (Tyr26 on S1). Thus, at 0 mV, and in the presence of cAMP,

we would identify an intermediate or an open state in such mutants, because the equilibrium is shifted from the closed towards the open state. We highlighted this better in the manuscript (lines 89-90, 146-149, 185-188, 245-248, 351-352, 368-370).

2. The author's suggest that longer mean open times in R120A vs Y26F are why open conformations were only observed for R120A. However, very little of the raw open time distributions are shown, and it would be nice to see more of the data that is summarized in Fig. 4F (for example, expand Fig. 7F to include all three channels and perhaps also a voltage closer to 0 mV to better correspond with the structures).

Thank you for the suggestion. We now included longer recordings for all three channels at +25 mV and the corresponding dwell time histograms with fits to a single component log probability function in Supplementary Fig. S8.

Also, in Fig. 4B-E it appears that compact conformations are also longer lived in R120A. Is this true?

We assume that the compaction that you refer to is the compaction in overall protein height, but it is the expansion of the CNBDs in the open state, which correlates to the decrease in FRET efficiency. Unfortunately, we do not have sufficiently long recordings to extract enough events and reliably determine the dwell times in each FRET state for each protein construct. With respect to the lifetime of the low FRET efficiency state, even for SthK R120A, these transitions to low FRET efficiency are the minority of events observed (~16 % in SthK R120A, Supplementary Table S4).

Minor Comments:

1. Lines 293-294: I don't understand why there are not sufficient closed events for accurate closed duration distributions. The data in Figs. 1D and S7D suggest otherwise.

We apologize for not being clear enough. The closed dwell times range from very short to very long and are fit with at least 3 components (Rheinberger et al., 2018). As a consequence, we need many events for each category, which requires upwards of 10 min of recording in each voltage and ligand condition, which is technically difficult to achieve with horizontal lipid bilayer recordings. Because of this, we have apparent heterogeneity in these components even for only one voltage/ligand condition, so comparisons between these different conditions in WT are already difficult, leave alone comparing different mutants. We updated the text to make this clearer.

Lines 318-321: "The closed dwell time distributions generally are heterogenous, contain several exponential components, ranging from seconds to microseconds, and have too few events per time bin for accurate fitting and comparisons, and we did not analyze them further."

2. Lines 253-255: Contrary to what is stated here, Fig. 8A suggests that C153A reduces the P_o of R120A in particular.

Indeed, SthK C153A R120A shows lower P_o values than SthK R120A. However, the open probabilities for both Y26F and R120A on the background of C153A are still higher than C153A alone. We improved the description in the main text to read.

Lines 275-276: “Although differences are less pronounced in the double-mutants, their open probabilities are still higher than for SthK C153A (Supplementary Fig S8A,E).”

3. Line 210: Define "slight S4 movement".

Thank you. We improved our writing (lines 226-228): “In contrast, in activated, pre-open SthK Y26F, S4 shows only a minimal displacement which is limited to the last helical turn and does not allow expansion of S6 (Supplementary Fig. S7E). This displacement is, however, accompanied by a repositioning of the S4-S5 loop, ...”

Marcel Goldschen-Ohm

Reviewer #3 (Remarks to the Author):

This manuscript by Gao et al. aims to determine the structures of intermediate states during SthK channel opening. SthK is a prokaryotic CNG channel with properties and structures common with vertebrate CNG channels, which have important physiological functions and whose dysfunction causes severe visual disorders. Structures of closed and fully open CNG channels have been obtained, but numerous questions remain to be elucidated on the gating process. Taking advantage of a partial agonist and a modest voltage dependence of SthK gating, and by using two gain-of-function mutations (Y20F and R120A) in the VSD, the authors stabilized and obtained the cryo-EM structures of several intermediate gating states of SthK. The authors then used sm-FRET to verify the existence of multiple states and single-channel analysis to correlate the structures to functional behaviors. This work successfully captures, for the first time, various transition states of a CNG channel en route from close to open, providing new mechanistic insights into allosteric gating of CNG channels and highlighting an inhibitory role of the VSD in the gating process. The methods used in this study could also be used to investigate other channels with similar gating mechanisms.

The experiments are well designed and executed, the data are convincing, the results are clear and interesting, and the paper is well written.

However, an important piece of information is missing: the structures of the apo state of both Y26F and R120A mutant channels. It is possible, or even highly likely, that these structures are identical to that of apo WT SthK. Still, it would be valuable to show that this is indeed the case. There is, however, a possibility that some mutant channels may display some global or local conformational changes (e.g., in the VSD, S4-S5 linker, S5 or S6) that are caused by the mutations and that can be captured by cryo-EM. It would be interesting, and indeed, important to examine this possibility. In this context, it is useful to note that a single amino acid mutation in the *C. elegans* CNG channel TAX-4 causes the channel to open and produces an open-state structure in the absence of cyclic nucleotides (Zheng et al., Communications Biology, 2022).

The reviewer raises an excellent point. We indeed considered collecting data for apo SthK Y26F and apo SthK R120A at the beginning of this project to test exactly that: whether our “mutant channels display some local or global conformational change that are caused by the mutations and could be captured by cryoEM”. However, we realized that our data contain these controls

already and spending the additional effort and funds on obtaining these structures was not justified for the following reasons:

First, it is important to note that the structures of WT SthK in apo and cAMP-bound conditions are identical and in a closed state. In addition, our cAMP-bound Y26F and R120A closed structures are identical to the cAMP-bound WT SthK in closed state. To show this more clearly, we now added Supplementary Fig S6C. If there were conformational changes in the apo state of SthK Y26F or R120A, these changes should also be observed in the cAMP-bound, closed mutant structures. However, we do not see changes in the closed states of the mutants and only observe conformational changes in the TMs when the CNBDs are in the activated state.

Second, for both mutants, SthK Y26F and SthK R120A, we identified only one closed state (defined by the majority of particles) which, furthermore, is identical to WT SthK in the apo state (and cAMP-bound state). This **indicates that the mutations alone do not induce structural changes**. We realized that we did not explicitly spell out the percentages of particles in the various states (these numbers were given in the processing schemes only (Supplementary Fig. S2 and S4)) and we did not sufficiently stress the identity between the closed mutant structures and the WT structures. We rectified this issue in the revised text (lines: 89, 144-148, 182-188, 199-200, 245-248, 351-352, 369). In summary the numbers are:

Protein/ state	Number of particles	%
SthK Y26F, closed	1112621	~91
SthK Y26F, pre-open, activated	109647	~9
SthK R120A, closed	799349	~76
SthK R120A, O1	50037	~5
SthK R120A, O2	149792	~14
SthK R120A, O3	46629	~5

These distributions fit very well with our electrophysiological data. In our original cryo-EM study of SthK (Rheinberger et al., 2018), WT SthK was detected only in a closed state conformation and the open probability P_o of WT SthK is below 0.1 at 0 mV. SthK Y26F shows an increased P_o and about 10 % of the particles adopt the pre-open, activated state. This observation is even more pronounced for SthK R120A, which shows the highest P_o and about 25 % of the particles fall into the three, structurally resolved open states.

Third, our electrophysiology data indicate that the mutant SthK channels studied here do not open in the absence of cAMP. Our single channel recordings directly show that in the absence of cAMP both mutant channels Y26F and R120A show zero activity (Supplementary Fig. S8D). This is in stark contrast to the example the reviewer raised where the mutant channel opens in the absence of ligands (Zheng et al., 2022). In agreement with this, sm-FRET revealed the same conformational distribution in the absence of cAMP for WT SthK and both mutant channels (Supplementary Fig. S8B). Both electrophysiology and smFRET strongly suggest that the apo WT and mutant channels are closed and that they visit similar closed states.

Taken these considerations together, we argue that we already have sufficient evidence that the mutations themselves do not lead to spontaneous channel openings or to local and/or propagated rearrangements in the structures. We hope the reviewer agrees.

Nevertheless, we think that the reviewer raised a legitimate point. We now highlight these thoughts better throughout the revised manuscript, where we also pointed out the identical structures of the closed mutants and WT, and have an additional supplemental figure highlighting the lack of local or global changes introduced by the mutations in the closed state, and the large number of particles in these closed conformations (lines: 89, 144-148, 182-188, 199-200, 245-248, 351-352, 369, Supplementary Fig. S6).

Minor comments/questions:

1. It would be helpful to zoom in around F26 and A120 and show the density maps and model fit to see the loss of interactions engaged by these residues and the local conformational changes caused by the mutations.

Thank you for this suggestion. We now prepared a supplementary figure showing the local environment of both positions and the absence of any conformational changes induced by the mutations. Furthermore, we added RMSD plots for both mutants which show, that the closed states are virtually identical to the closed state of WT SthK (Supplementary Fig. S6, executive RMSDs between WT SthK and SthK Y26F is 0.4 Å, and SthK R120A is 0.7 Å).

2. A displacement of the S4-S5 linker appears to be crucial for the pre-open state of Y26F. It would be helpful to show the interactions and Y26F mutation-induced changes, if there are any, between the S4-S5 linker and the C-linker.

This is another good point raised by the reviewer. In fact, we did not observe any mutation-induced conformational changes in the closed states of SthK Y26F or R120A (new Supplementary Fig. S6). This is exactly what we intended to achieve: introduce mutations that abolish certain interactions of S4 with surrounding helices from the VSD to alter the relative stabilities between the closed and open state without changing the structures of the protein in the resting state. Reviewer 2 had a very similar comment and we have changed the manuscript at various points to make it more clear that the closed states between WT and the two mutants are identical. (lines 89-90, 146-149, 185-188, 245-248, 351-352, 368-370)

3. In eukaryotic CNG channels, including TAX-4, CNGA1, CNGA1/CNGB1 and CNGB3/CNGB3, two hydrophobic residues form a gate in the central cavity, and this gate has been called a cavity gate or central gate. The authors call the gate in SthK the “intracellular bundle crossing gate”. Is the gate in SthK formed solely by I215 in SthK? Although it may be a matter of semantics, it is notable that I215 is 2-3 alpha-helical turns up than the “inner” or “bundle” gates in many other channels. A brief discussion about the similarity/dissimilarity of the activation gate in eukaryotic CNG channels and SthK may be useful.

Thank you for this comment. The reviewer is absolutely correct, and we agree that the bundle-crossing gate in KcsA, for instance, and the cavity/central gate in CNG channels are not exactly the same. This is also true for SthK. We now point out more clearly that Ile215 is the main constriction in SthK and is similar to the central gate of CNG channels. In addition to the central gate, SthK also has a constriction (less pronounced) in the bundle-crossing region. Rotation of the C-linker/CNBD domains generates strain in the bundle crossing region at the end of S6. Opening of the bundle crossing is accomplished by an outwards rotation of S6 which is propagated to the gate formed by Ile215. In activated, pre-open SthK Y26F the bundle-crossing constriction is destabilized, but sterically still closed, with S6 not rotated outwards, and accordingly, the I215 gate is also still closed. We updated the text as follows.

Lines 167-172: “However, this expansion is limited to the very bottom of the intracellular entry to the pore (the last helical turn of S6) and is insufficient to generate an outwards rotation of S6. Thus, the lumen above Ser223, which is similar to the central gate found in eukaryotic CNG channels, is not changed as compared to the closed state. Especially, Ile215, which forms the main constriction in SthK and other CNG channels, still pinches the pore sterically shut below the selectivity filter (Fig. 3A,C) (Rheinberger et al., 2018; Xue et al., 2021).”

Lines 203-205: “Along with the S6 backbone translation and rotation away from the pore axis, the side chains of Ile215, which form the main constriction in the closed state similar to the cavity gate in CNG channels, also swing away from the pore ...”

Lines 231-232: “... allowing for only negligible movements of S6 below the central gate level (i.e. below Ile215, Fig. 2C and Fig. 3E).”

4. In this study, SthK nanodiscs were reconstituted in DOPC:POPG:Cardiolipin. In previous studies by the same group, only POPG was used. Any particular reason(s) for this change of lipid composition?

In our previous studies, we indeed used POPG alone in nanodiscs, since it yielded good cryo-EM samples. Since then, we succeeded to also produce good samples in the same exact lipid composition as used for our functional assays (5:3:2 DOPC:POPG:CL) and we observed no structural changes. However, in considering the lipid composition of the nanodiscs for this structural study, we wanted to apply best practices and be as faithful as possible to the functional data, since central to the present work is the correlation between protein structure and protein function. We now spell this out in the results.

Lines 141-143: “... we reconstituted both proteins into lipid nanodiscs using the same lipid composition as in our functional experiments (DOPC:POPG:Cardiolipin, 5:3:2) for cryo-EM studies.”

5. Some parameters in structure refinement and validation seem to have room for improvement, including the clashscore, Ramachandran outliers, and preferred Ramachandran ratio.

Thank you for this observation. We could indeed do better. We now further improved the models for pre-open, activated SthK Y26F, closed SthK R120A, and O1 SthK R120A (see Supplementary Table S2 and S3), where the clashscores indeed were high. With this improvement, also bond lengths and angles improved, as well as the preferred Ramachandran ratios. The structures had no Ramachandran outliers to begin with, which did not change.

6. There is an error in the citation of Carrasquel-Ursulaez et al.

Thank you for the catch! The reference is now corrected.

References

- Dai, G., T.K. Aman, F. DiMaio, and W.N. Zagotta. 2019. The HCN channel voltage sensor undergoes a large downward motion during hyperpolarization. *Nat Struct Mol Biol.* 26:686-694.
- Kasimova, M.A., D. Tewari, J.B. Cowgill, W.C. Ursuleaz, J.L. Lin, L. Delemotte, and B. Chanda. 2019. Helix breaking transition in the S4 of HCN channel is critical for hyperpolarization-dependent gating. *Elife.* 8.
- Lee, C.H., and R. MacKinnon. 2017. Structures of the Human HCN1 Hyperpolarization-Activated Channel. *Cell.* 168:111-120.e111.
- Lee, C.H., and R. MacKinnon. 2019. Voltage Sensor Movements during Hyperpolarization in the HCN Channel. *Cell.* 179:1582-1589.e1587.
- Rheinberger, J., X. Gao, P.A. Schmidpeter, and C.M. Nimigean. 2018. Ligand discrimination and gating in cyclic nucleotide-gated ion channels from apo and partial agonist-bound cryo-EM structures. *Elife.* 7:e39775.
- Roy, R., S. Hohng, and T. Ha. 2008. A practical guide to single-molecule FRET. *Nat Methods.* 5:507-516.
- Ryu, S., and G. Yellen. 2012. Charge movement in gating-locked HCN channels reveals weak coupling of voltage sensors and gate. *J Gen Physiol.* 140:469-479.
- Schmidpeter, P.A.M., X. Gao, V. Uphadyay, J. Rheinberger, and C.M. Nimigean. 2018. Ligand binding and activation properties of the purified bacterial cyclic nucleotide-gated channel SthK. *J Gen Physiol.* 150:821-834.
- Shuang, B., D. Cooper, J.N. Taylor, L. Kisley, J. Chen, W. Wang, C.B. Li, T. Komatsuzaki, and C.F. Landes. 2014. Fast Step Transition and State Identification (STaSI) for Discrete Single-Molecule Data Analysis. *J Phys Chem Lett.* 5:3157-3161.
- Xue, J., Y. Han, W. Zeng, Y. Wang, and Y. Jiang. 2021. Structural mechanisms of gating and selectivity of human rod CNGA1 channel. *Neuron.* 109:1302-1313.e1304.
- Zagotta, W.N., T. Hoshi, J. Dittman, and R.W. Aldrich. 1994. Shaker potassium channel gating. II: Transitions in the activation pathway. *J Gen Physiol.* 103:279-319.
- Zheng, X., H. Li, Z. Hu, D. Su, and J. Yang. 2022. Structural and functional characterization of an achromatopsia-associated mutation in a phototransduction channel. *Commun Biol.* 5:190.

Reviewers' Comments:

Reviewer #1:

Remarks to the Author:

Major concerns have been addressed satisfactorily and the paper has been improved substantially. One correction is needed in Page 4, line 86: "SthK ($z \sim 0.8$) to Shaker ($z \sim 4$ (Zagotta et al., 1994)) and HCN ($z \sim 1$ (Ryu and Yellen, 2012))". Using the Boltzmann's fit of P_o versus voltage, the z value obtained for sea urchin spHCN channel is around 2.7, not 1. Ryu and Yellen, 2012 paper plotted gating currents (rather than P_o) versus voltage. The z value of HCN2 is > 5 , and z of HCN1 > 4 . Please modify this and perhaps cite papers: Gauss et al., Nature, 1998; Kusch et al., Neuron, 2010 or other papers from the Siegelbaum lab.

Reviewer #2:

Remarks to the Author:

The authors have sufficiently addressed all of the comments.

Reviewer #3:

Remarks to the Author:

The authors have adequately addressed my concerns and questions. I have no further comments.